

# Insights into the prediction uncertainty of machine-learning-based digital soil mapping through a local attribution approach

Jeremy Rohmer[1], Stephane Belbeze[1], Dominique Guyonnet[1]

[1]BRGM, 3 av. C. Guillemin - 45060 Orléans Cedex 2, France

*Correspondence to*: Jeremy Rohmer (j.rohmer@brgm.fr)

**Abstract.** Machine learning (ML) models have become key ingredients for digital soil mapping. To improve the interpretability of their prediction, diagnostic tools have been developed like the widely used local attribution approach known as 'SHAP' (SHapley Additive exPlanation). However, the analysis of the prediction is only one part of the problem and there is an interest in getting deeper insights into the drivers of the prediction uncertainty as well, i.e. to explain why the ML model

is confident, given the set of chosen covariates' values (in addition to why the ML model delivered some particular results). We show in this study how to apply SHAP to the local prediction uncertainty estimates for a case of urban soil pollution, namely the presence of petroleum hydrocarbon in soil at Toulouse (France), which poses a health risk via vapour intrusion into buildings, direct soil ingestion or groundwater contamination. To alleviate the computational burden posed by the multiple covariates (typically >10) and by the large number of grid points on the map (typically over several 10,000s), we propose to

rely on an approach that combines screening analysis (to filter out non-influential covariates) and grouping of dependent covariates by means of generic kernel-based dependence measures. Our results show evidence that the drivers of the prediction best estimate are not necessarily the ones that drive the confidence in these predictions, hence justifying that decisions regarding data collection and covariates' characterisation as well as communication of the results should be made accordingly.

## 1 Introduction

Maps of soil physical properties like, cation exchange capacity, pH, soil organic content, hydraulic properties, etc., or chemical properties like concentrations of heavy metals (arsenic, mercury, etc.) or radionuclide (Caesium 137, Plutonium-239+240) or target-oriented indicators like erodibility, soil compaction (see e.g. Panagos et al., 2022) are keys to multiple studies as diverse as pollution assessment, urban planning, construction design, etc. Over the recent years, this mapping has witnessed many advances to improve spatial prediction (in relation to the domain of digital soil mapping (DSM), McBratney et al., 2003) with

the developments of methods and approaches either based on the geostatistical framework (Chiles and Desassis, 2018) or on machine learning (denoted ML) techniques (Wadoux et al. 2020).

Beyond spatial prediction, the question of uncertainty in the spatial prediction has emerged as a key challenge (Heuvelink and Webster, 2022: Sect. 4). Historically, this question has been addressed with kriging (see e.g., Veronesi & Schillaci, 2019 for a discussion for DSM), and ML techniques have increasingly been used (or adapted) for this goal, for instance, based on the




popular quantile random forest model (e.g., Vaysse & Lagacherie, 2017), among others, for producing soil information
worldwide in the SoilGrids 2.0 database together with uncertain information (Poggio et al., 2021). Along these developments,
improvements of validation procedures have been proposed (Schmidinger & Heuvelink 2023) together with tools for assessing
the prediction error and transferability (Ludwig et al., 2023).

Quantifying the uncertainty is however only one part of the problem and there is an interest in gaining deeper insights into the
influence of each covariate on the overall prediction uncertainty. This is the objective of global sensitivity analysis (Saltelli et
al., 2008), which can be conducted within two settings, either "Factor Fixing" to identify non-influential covariates, or "Factor
Prioritization" to rank in terms of importance the covariates. The expected results can be of different types: the former setting
provides justifications for simplifying the spatial predictive model (by removing the non-influential covariates), whereas the
latter setting provides justifications for prioritizing future characterization efforts (by focusing on the most important
variables). In DSM, this question has been addressed with the tools of variance-based global sensitivity analysis (i.e. the Sobol'
indices as implemented by Varella et al., 2010) or with variable importance scores together with potentially recursive feature
elimination procedures (as implemented by Poggio et al. (2021) and Meyer et al., 2018). Both approaches provide a global
answer to the problem of sensitivity analysis, i.e. by exploring the influence over the whole range of variation of the covariates.
However, at a local scale, these methods provide any information for a prediction at a certain spatial location.

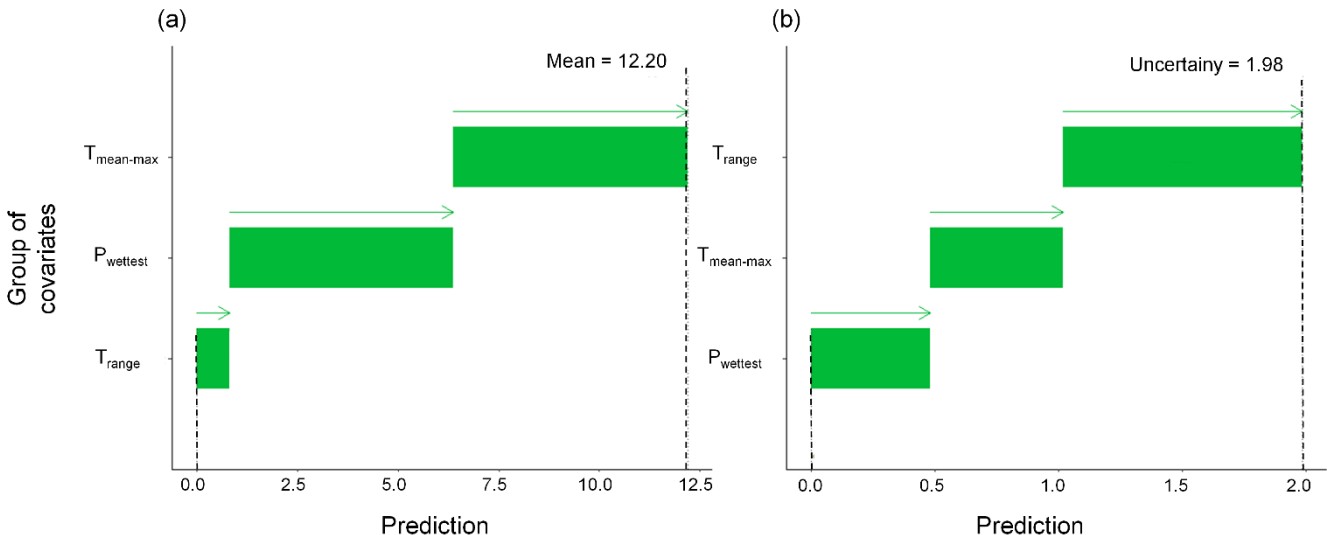


**Figure 1: Example of SHAP-based decomposition of the prediction best estimate (modelled by the conditional mean of a random forest model, panel a) and of the prediction uncertainty measured by the inter-quartile width (estimated via a quantile random forest model, panel b) for the variable of interest of the synthetic test case (fully described in Sect. 2.1) at a certain location of the study area. Each horizontal bar provides the contribution to the prediction (indicated by the vertical dotted line) of the considered**
**covariates (indicated on the vertical axis) that correspond to the mean diurnal range $T_{range}$, to the group of covariates including the maximum and mean temperature of the warmest quarter of the year $T_{mean-max}$, and to the precipitation of wettest month $P_{wettest}$. Note the differences in the ordering of the groups, which indicate that the contributors to the mean and to the uncertainty estimate differ.**



Recently, an alternative local approach has been proposed by relying on a popular method from the domain of interpretable

machine learning (Molnar, 2022) based on the Shapley values (Shapley, 1953). This method has shown promising results to attribute contributions of each covariate to any spatial prediction (Padarian et al., 2020; Wadoux et al., 2023; Wadoux and Molnar, 2022).

The application of the Shapley values for DSM is however focused on the prediction best estimate and little information is provided on the local prediction uncertainty. To fill this gap, our objective is twofold. First, motivated by a case of pollution

concentration mapping in the city of Toulouse, France (Belbeze et al., 2019), we investigate how to use the Shapley values to decompose the local uncertainty (either measured by an inter-quantile width or by a variance estimate). The benefit is to provide evidence that the drivers of the prediction best estimate are not necessarily the ones that drive the confidence in the predictions, i.e. decision in terms of data collection and covariates' characterisation may differ depending on the target, the prediction best estimate or the confidence/uncertainty. Communication on the results should also be adapted accordingly.

Figure 1 illustrates the type of result that can be derived with the approach. In this example (based on the synthetic test case fully detailed in Sect. 2.1), the mean prediction (best estimate, left panel) of the variable of interest at a certain location does not have the same contributors as the local uncertainty measured by the inter-quartile width (right panel). The group of covariates including the maximum and mean temperature of warmest quarter of the year (named $T_{mean-max}$) is respectively identified as the first and second most important contributor. The identification of the least influential group of covariates also

differs across both cases: this is illustrated with the mean diurnal range $T_{range}$ which has little impact on the prediction result, but strongly influences the confidence in the result.

To reach this type of result, two main practical difficulties have to be addressed. This is our second contribution, which focuses on: (1) the high computational burden of the estimation of the Shapley values (as discussed by Wadoux et al., 2023 for DSM) despite some advances in the machine-learning domain like the kernel SHAP method of Lundberg and Lee (2017), and (2) the

influence of covariates' dependence (Aass et al., 2021). To alleviate both problems, we rely on the simple-but-efficient approach of Jullum et al. (2021) based on grouping of dependent covariates.

The paper is organised as follows. We first describe the two application cases that motivated this study. In Sect. 3, we provide further details on the statistical methods that are used to estimate the local explanations of the prediction uncertainty. In Sect. 4, we apply the methods and provide an in-depth analysis of the differences in the drivers of prediction best estimate and

uncertainty. In Sect. 5, we discuss the practical implications of the proposed procedure and its limitations.

## 2 Case study

In this section, we describe the two application cases that are used, namely a synthetic (Sect. 2.1) and a real test case (Sect. 2.2), to showcase the procedure for local uncertainty decomposition described in Sect. 3.





## 2.1 Synthetic test case

The first test case is synthetic. It aims to predict a virtual species suitability surface denoted *y* over central Europe (Figure 2). It is calculated based on six bioclimatic covariates, defined in Table 1 (with prior normalisation between 0 and 1) extracted from the Worldclim dataset (available at: www.worldclim.org/data/bioclim.html) as follows:

$$y(s) = 10 \times T_{\text{range}}(s) + 5 \times T_{\text{max}}(s) + 5 \times T_{\text{mean}}(s) + 5 \times P_{\text{wettest}}(s) + 10^{-4} . P_{\text{driest}}(s) + 10^{-4} . P_{\text{coldest}}(s), \quad (1)$$


By construction, this model has two characteristics that are used here to validate the proposed methods described in Sect. 3.4: (1) the two last covariates have negligible influence; (2) the covariates $T_{\text{max}}$ and $T_{\text{mean}}$ are strongly dependent. The dataset is based on the vignette of the R package *CAST* available at: https://hannameyer.github.io/CAST/articles/cast02-AOA-tutorial.html. A series of 50 "virtual" soil samples are randomly extracted (highlighted by square-like markers in Fig. 2) across

the study area.

**Table 1. Description of the covariates for the synthetic test case.**

| Covariate | Unit | Description | Identifier in Worldclim dataset |
|---|---|---|---|
| $T_{\text{range}}$ | °C | Mean diurnal range i.e. mean of monthly (max temperature - min temperature)). | *Bio2* |
| $T_{\text{max}}$ | °C | Max temperature of warmest month | *Bio5* |
| $T_{\text{mean}}$ | °C | Mean temperature of warmest quarter | *Bio10* |
| $P_{\text{wettest}}$ | mm | Precipitation of wettest month | *Bio13* |
| $P_{\text{driest}}$ | mm | Precipitation of Driest Month | *Bio14* |
| $P_{\text{coldest}}$ | mm | Precipitation of Coldest Quarter | *Bio19* |





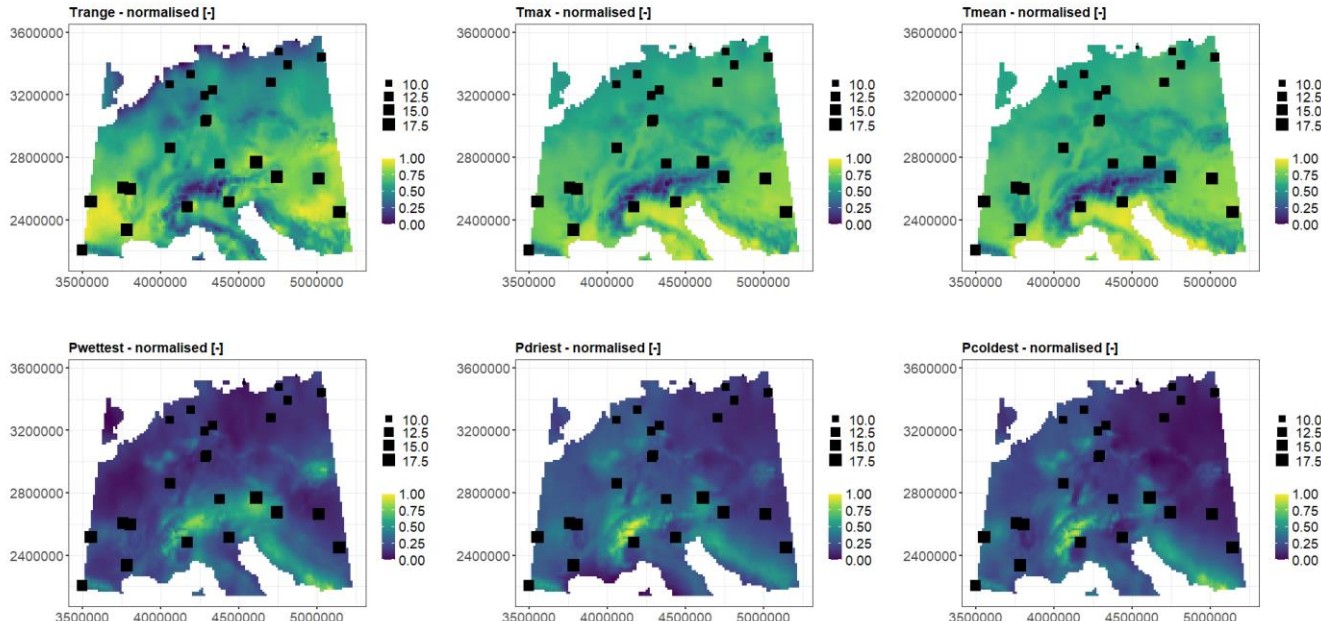

**Figure 2: Covariates (with prior normalisation between 0 and 1) used in the synthetic test case (see Table 1 for a detailed description).**
**Spatial location of the 50 soil samples are indicated by square-like markers. The size of the squares is proportional to the synthetic variable calculated from the covariates based on Eq. (1).**

## 2.2 Real test case

The real case is focused on the DSM performed for the prediction of total petroleum hydrocarbon (C10-C4) concentration over the city of Toulouse (located in the South West of France), as part of the definition of urban soil geochemical backgrounds
(see a comprehensive review by Belbeze et al. (2023)). The presence of petroleum hydrocarbon (in this case, from multiple sources such as road and air traffic, industrial emissions, residential heating, etc.) may inhibit several soil functions and hence prevent the delivery of associated ecosystem services (Adhikari and Hartemink, 2016). A primary soil function that may be jeopardised is the soil ability to provide a platform for human activities in a risk-free environment. Petroleum hydrocarbons in soil generate a risk for human health via several pathways; e.g., direct soil ingestion (a particularly sensitive pathway for young
children); vapour intrusion into buildings (and exposure through respiration) or contamination of groundwater used for drinking water purposes. Noteworthy, our study uses the data of this case to illustrate and discuss the applicability of the proposed approach, and is not meant to supplement the results of Belbeze et al. (2019).



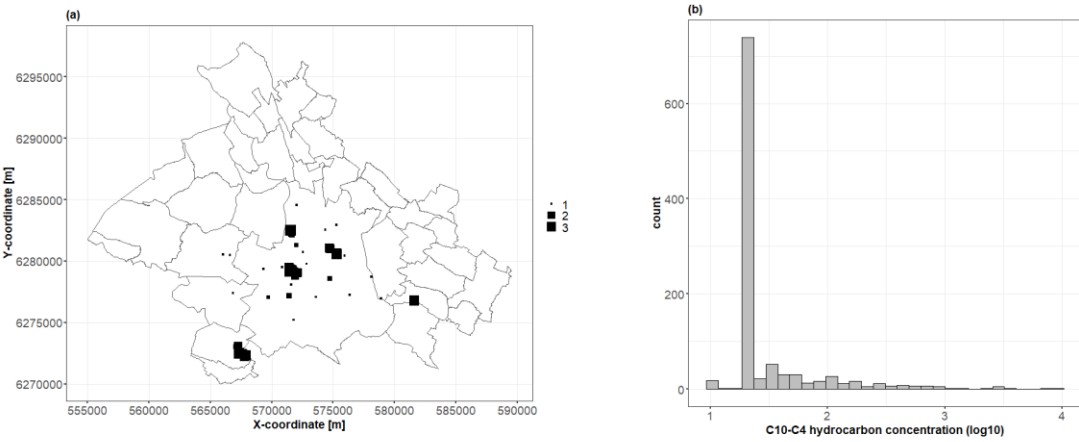

**Figure 3: (a) Spatial location of the 1,043 soil samples (square-like markers) across Toulouse city. The size of the squares is**
**proportional to the logarithm (base 10) of the C10-C4 hydrocarbon concentration (expressed in mg/kg). (b) Histogram of the**
**logarithm (base 10) of the C10-C4 hydrocarbon concentration (expressed in mg/kg).**

We use 1,043 soil samples (collated over a depth interval [0, 2m]) to analyse the logarithm (base 10) of the C10-C4
hydrocarbon concentrations, expressed in mg/kg (Fig. 3). We aim at predicting the concentration over the whole Toulouse city
using a fine grid of spatial locations (one point every 100m, i.e. >45,000 grid points) using the covariates described in Table
2. Figures 4 and 5 depict the spatial distribution of the considered covariates, of continuous and categorical type respectively.

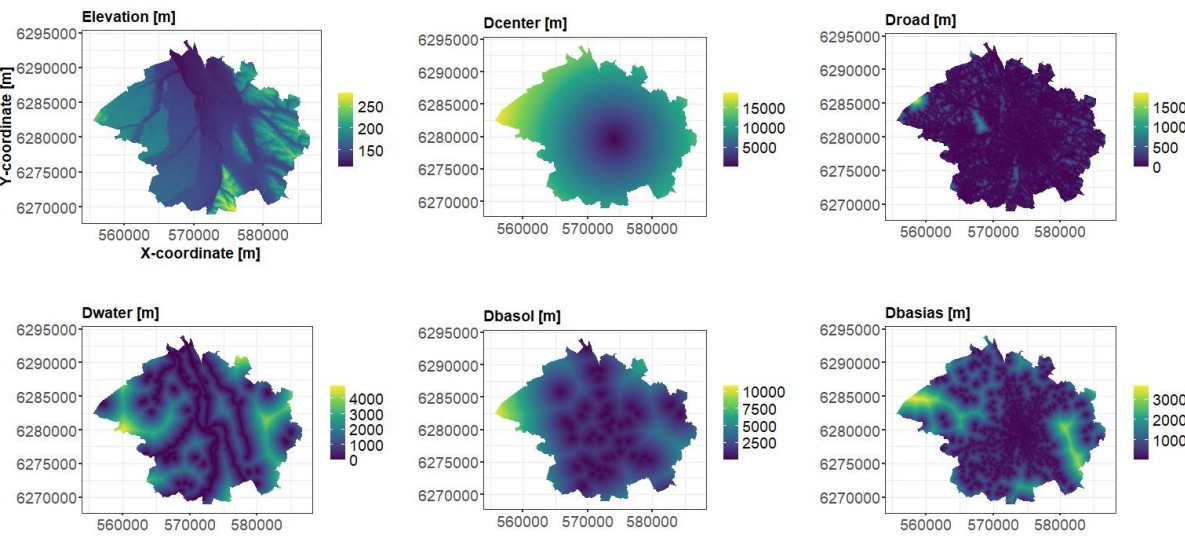

**Figure 4: Covariates of continuous type over Toulouse city (see Table 2 for a detailed description).**



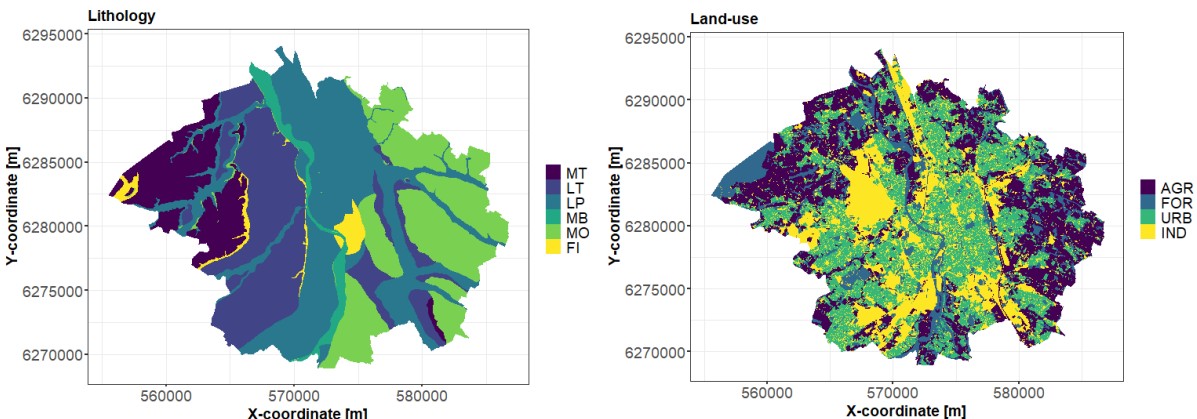

**Figure 5: Covariates of categorical type over Toulouse city. Left. Lithology (MT: Medium Terrace alluviums, LT: Low Terrace**
**alluviums, LP: Low Plain Alluviums, MB: Major River Bed alluviums, MO: Molasses, FI: Fill materials); Right. Land use (AGR:**
**Agriculture, FOR: Forests and grasslands, IND: Industrial and commercial economic activities), see Table 2 for a detailed**
**description.**

In addition to these covariates, we follow the approach proposed by Behrens et al. (2018) to better account for spatial
dependence: we also consider seven additional covariates, namely the two geographical coordinates, $X$, and $Y$, five
geographical covariates that correspond to the distances to the south-eastern, the north-eastern, the south-western and the north-
western corners of a rectangle around the city (denoted $D_{se}$, $D_{ne}$, $D_{sw}$, $D_{nw}$), and the distance to the centre location of this
rectangle (denoted $D_{middle}$). A total of 15 covariates are considered.

**Table 2. Description of the covariates of the real test case.**

| Covariate | Unit | Description | Source |
|---|---|---|---|
| Elevation | [m] | Digital Elevation model post-processed from LIDAR data and gridded at a 10 m x 10 m resolution | Based on the processing detailed in (Belbeze et al., 2022) |
| Lithology | - | 8 categories based on the grouping of Belbeze et al., (2019) | Based on CHARM database available at https://www.data.gouv.fr/fr/datasets/cartes-geologiques-departementales-a-1-50-000-bd-charm-50/ |



| Land-Use | - | 4 categories based on the grouping of Belbeze et al., (2019) | Based on Copernicus (2012) available at https://doi.org/10.2909/debc1869-a4a2-4611-ae95-daeefce23490 |
|---|---|---|---|
| $D_{basias}$ | [m] | Distance to industrial sites (abandoned or in activity) potentially at the origin of pollution | BASIAS database (Leprond (2013)) available at https://www.data.gouv.fr/en/datasets/inventaire-des-sites-pollues/ |
| $D_{basol}$ | [m] | Distance to polluted (potentially) sites | BASOL database available at https://www.data.gouv.fr/fr/datasets/base-des-sols-pollues/ |
| $D_{road}$ | [m] | Distance to the closest roads | Based on the processing detailed in (Belbeze et al., 2019) |
| $D_{water}$ | [m] | Distance to the closest rivers | Based on the processing detailed in (Belbeze et al., 2019) |
| $D_{center}$ | [m] | Distance to the city centre | Based on the processing detailed in (Belbeze et al., 2019) |
| $X$ and $Y$ coordinate | [m] | Geographical coordinate in the coordinate reference system of France, Lambert 93 | - |
| $D_{se}$, $D_{ne}$, $D_{sw}$, $D_{nw}$, $D_{middle}$ | [m] | Distance to the south-eastern, the north-eastern, the south-western and the north-western corners of a rectangle around the city, and the distance to the centre location of this rectangle | Based on the approach of Behrens et al. (2018) |

## 3 Methods

In this section, we first describe the different steps of the proposed procedure (Sect. 3.1). The details of the methods at each step are described in the subsequent sections (Sect. 3.2-3.4).





### 3.1 Overall procedure

Let us first consider that the value of the variable of interest $y(s)$ (soil property, pollutant concentration, etc.) at a given spatial location $s$ is related to $d$ covariates $\mathbf{x}(s) = \{x_1(s), x_2(s), \ldots, x_d(s)\}$. The mathematical relationship is modelled by a ML model denoted $f(.)$, i.e. $y(s) = f(\mathbf{x}(s))$. The ML model also provides a measure of uncertainty (denoted $u(s)$) on this prediction that is related to $\mathbf{x}(s)$ through the function $g(.)$ (that may differ from $f(.)$). In this study, we focus on the random forest model (denoted RF) used for regression and on its quantile regression variant (models named qRF), because this ML model has

proven to be very efficient in multiple studies for DSM as outlined in the introduction. Further technical details are provided in Sect. 3.2.

Our objective is to decompose $u(s)$ at a given spatial location $s$ as a sum of the covariates' $\mu_{i=1,\ldots,d}(s)$ specific to the values of covariates $\mathbf{x}(s)$ within the setting of the additive "feature attribution approach" (as defined by Lundberg and Lee (2017): Sect. 2) as follows:

$$u(s) = g(\mathbf{x}(s)) = \mu_0 + \sum_{j=1}^{p} \mu_j(s),\tag{2}$$

where $\mu_0$ (named *base value*) is a constant value (see definition in Sect. 3.3). This decomposition can also be applied to $f(.)$ as done in previous studies as indicated in the introduction.

It is important to note that Eq. (2) does not aim to linearise $g(.)$, but to compute the contribution of each covariate to the particular prediction uncertainty value $g(\mathbf{x}(s))$. This means that the decomposition provides insights into the influence of the

particular instance of the covariates $\mathbf{x}(s)$ relative to $g(\mathbf{x}(s))$: (1) the absolute value of $\mu(s)$ informs on the magnitude of the influence at the location $s$ directly expressed in physical units, which eases the interpretation; (2) the sign of $\mu(s)$ indicates the direction of the contribution, i.e. whether the considered covariate pushes the prediction higher or lower than the base value $\mu_0(s)$. Both aspects are outlined in Fig. 1; the width of the horizontal bar and the arrow being an indicator of (1) and (2) respectively. In order to quantify $\mu(s)$ in Eq. 2, we will rely on the approach named SHapley Additive exPlanation (SHAP)

developed by Lundberg and Lee (2017) described in Sect. 3.3.



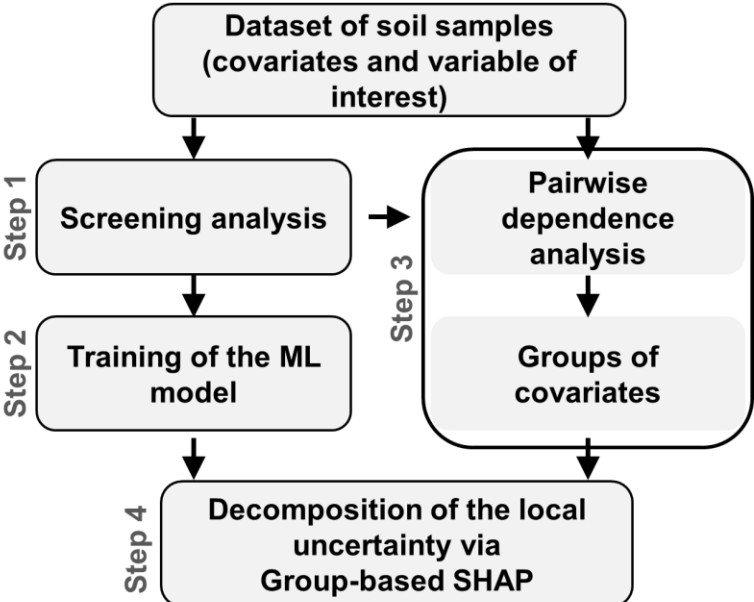

**Figure 6: Workflow of the proposed procedure for decomposing the prediction uncertainty into a sum of contributions linked to each group of covariates.**

The different steps of the proposed approach named group-based SHAP (schematically represented in Fig. 5; see implementation details in Appendix A) are as follows.

*Step 1* aims at identifying which among the $d$ covariates, have negligible influence on *Y*. This screening analysis aims to decrease the number of considered covariates, which allows to reduce the computational burden of SHAP (as discussed in Sect. 3.3). We rely here on a hypothesis testing approach (see El Amri and Marrel (2021) and references therein) based on the

Hilbert-Schmidt independence criterion, denoted HSIC (Gretton et al., 2005) described in Sect. 3.4;

*Step 2* aims at building and training the RF models based on the dataset of soil samples together with the covariates' values.

*Step 3* aims at further decreasing the computational load of SHAP by identifying groups of covariates (as explained in Sect. 3.3) which share strong dependence. We rely on the *HSIC* measure (described in section 3.4) to quantify pairwise dependence between covariates, as this measure is generic (it applies to many types of dependence, i.e. linear, monotonic associate, or non-

linear, see a discussion by Song et al., 2022) and can be applied to mixed variables, i.e. continuous or categorical (as in our case described in Sect. 2).

*Step 4* aims at computing the Shapley values associated to each of the groups identified at *Step 3* in order to decompose the prediction uncertainty provided by the qRF model (trained at *Step 2*).



### 3.2 Quantile Random Forest

The Random Forest (RF) model, as introduced by Breiman (2001), is used here for regression. It is a non-parametric technique based on a combination (ensemble named "forest") of regression trees (Breiman et al. 1984). Each tree is constructed by relying on recursive partitioning, which aims at finding an optimal partition of the covariates' domain of variation by dividing it into L disjoint sets $R_1, …, R_L$ to have homogeneous $Y_i$ values in each set $R_{l=1,…,L}$ by minimizing a splitting criterion (e.g. based on the sum of squared errors, see Breiman et al. 1984) or when the number of observations in each partition reaches a minimal

number termed nodesize (denoted *ns*). To sum up, the RF model aggregates the different regression trees as follows: (1) random bootstrap sample from the training data and randomly select $m_{try}$ variables at each split; (2) construct $n_{tree}$ trees T($\alpha$), where $\alpha_t$ denotes the parameter vector based on which the $t^{th}$ tree is built; (3) aggregate the results from the prediction of each single tree to estimate the conditional mean of *Y* as follows:

$$Y(s) = f(\mathbf{x}(s)) = E(Y|X = \mathbf{x}(s)) = \sum_{j=1}^{n} w_j(\mathbf{x}(s))Y_j, \tag{3}$$

where $E(.)$ is the mathematical expectation, and the weights $w_j$ are defined as follows:

$$w_j(\mathbf{x}(s)) = \frac{\sum_{t=1}^{n_{tree}} w_t(\mathbf{x}(s), \alpha_t)}{n_{tree}}, \text{ with } w_j(\mathbf{x}(s), \alpha) = \frac{I_{\{X_i \in R_{l(x,\alpha)}\}}}{\#\{j: X_i \in R_{l(x,\alpha)}\}}, \tag{4}$$

where $I(A)$ is the indicator operator which equals 1 if *A* is true, 0 otherwise; $R_{l(x,\alpha)}$ is the partition of the tree model with parameter $\alpha$ which contains $x$.

The RF method is very flexible and can be adapted to predict quantiles. The quantile random forest (qRF) model was originally

developed by Meinshausen (2006), who proposed to estimate the conditional quantile $q_\tau(y|\mathbf{x})$ at level $\tau$ as

$$q_\tau(y|\mathbf{x}) = \inf(\mathbf{x}: F_{H|\mathbf{X}}(y|\mathbf{x}) \geq \tau), \tag{5}$$

where

$$F_{Y|\mathbf{X}}(y|\mathbf{x}) = \sum_{j=1}^{n} w_j(\mathbf{x})I_{\{Y_j \leq y\}}, \tag{6}$$

where the weight are calculated in the same manner as for the regression RF model.

The major difference with the formulation for regression RF is that the qRF model computes a weighed empirical CDF of *Y* for each partition instead of computing a weighed average value (as in Eqs. 3-4). To reflect the magnitude of the RF prediction uncertainty $u(s)$ at spatial location s, we use the inter-quantile half-width (denoted $IQW_\alpha$) at a given level $\alpha$ defined as follows:

$$u(s) = g(\mathbf{x}(s)) = IQW_\alpha = q^{\frac{1+\alpha}{2}}(y|\mathbf{x}(s)) - q^{\frac{1-\alpha}{2}}(y|\mathbf{x}(s)), \tag{7}$$

In particular, the inter-quartile width corresponds to $IQW_{\alpha=0.50}$

### 3.3 Shapley additive explanation


The SHAP approach relies on the Shapley value (Shapley, 1953), which is used in game theory to evaluate the "fair share" of a player in a cooperative game, i.e. it is used to fairly distribute the total gains to multiple players working cooperatively. It is a "fair" distribution in the sense that it is the only distribution satisfying some desirable properties (Efficiency, Symmetry,



Linearity, 'Dummy player', see proofs by Shapley, 1953, see Aas et al., 2021: Appendix A for a comprehensive interpretation
of these properties from a ML model perspective).

Formally, consider a cooperative game with $d$ players and let $D_S \subseteq D = \{1, \dots, d\}$ be a subset of $|D_S|$ players. Let us define a

real-valued function that maps a subset $D_S$ to its corresponding value val: $2^{D_S} \to \mathbb{R}$ and measures the total expected sum of

payoffs that the members of $D_S$ can obtain by cooperation. The gain that the i$^{\text{th}}$ player gets is defined by the Shapley value with

respect to val(.):

$$\mu_i(s) = \frac{1}{d} \sum_{D_S \subseteq D \setminus \{i\}} \binom{d-1}{|D_S|}^{-1} (\text{val}(D_S \cup \{i\}) - \text{val}(D_S)),$$   (8)

Equation 8 can be interpreted as a weighted mean over contribution function differences for all subsets $D_S$ of players not

containing player i. This approach can be translated for the ML-based prediction by viewing each covariate as a player, and

by defining the value function val(.) as the expected output of the ML model conditional on $\mathbf{x}_S^*$ i.e. when we only know the

values of the subset $D_S$ of inputs (Lundberg and Lee, 2017). This approach is flexible to any output of the ML model and can

correspond to the conditional mean of the RF model (Eq. 3) as well to the uncertainty measure computed with the qRF model

(Eq. 7).

Formally,

$$\text{val}(D_S) = \text{E}\big(h(\mathbf{x}) \big| \mathbf{x}_{D_S} = \mathbf{x}_{D_S}^*\big),$$   (9)

where $h(.)$ can either correspond to the conditional mean denoted $f(.)$, or the uncertainty estimate, denoted $g(.)$.

In this setting, the Shapley values can then be interpreted as the contribution of the considered covariate to the difference

between the prediction $h(\mathbf{x}^*)$ and the base value $\mu_0$. The latter can be defined as the value that would be predicted if we did

not know any covariates (Lundberg and Lee, 2017). In the application case, we are interested in the pollution prediction; in

this context, we choose $\mu_0 = 0$, which means that, if we did not know any covariates, no pollution is expected (and no

uncertainty). In this way, $\mu_i$ in Eq. 2 corresponds to the change in the expected model prediction if $f(.)$ is used (or in the

uncertainty if $g(.)$ is used) when conditioning on that covariate and explains how to depart from 0. By construction, if $\mu_i = 0$,

it indicates the absence of influence for the i$^{\text{th}}$ covariate (related to the 'dummy player' property of the method). In addition,

the sum of the inputs' contributions is guaranteed to be exactly $h(\mathbf{x}^*(s)) - \mu_0$ (related to the 'efficiency' property of the

method). Thus to ease the comparison between the different predictions across the study area, we analyse in the following a

scaled version of the Shapley absolute value, i.e. $|\mu(s)|/(h(\mathbf{x}^*(s)) - \mu_0)$ expressed in %.

In practice, the computation of the Shapley value may be demanding because Eq. (8) implies covering all subsets $D_S$ (which

grows exponentially with the number of factors denoted $d$, i.e. $2^d$), and Eq. (9) requires solving integrals, which are of

dimension 1 to $d-1$. When the SHAP approach is applied for a large number of spatial locations (in our case >45,000), this

computational complexity is demanding.



To further alleviate the computational burden, we rely on the group-based approach proposed by Jullum et al. (2021), which showed that Eq. 8 can be adapted for a group of covariates. Considering a partition $\boldsymbol{G} = \{G_1, G_2, \ldots, G_g\}$ of the covariate set $D$, the Shapley value for the i$^{\text{th}}$ group of covariates $G_i$ reads as follows:

$$\mu_{G_i}(s) = \frac{1}{g}\sum_{T \subseteq \boldsymbol{G}\backslash\{G_i\}} \binom{g-1}{|T|_g}^{-1} (\text{val}(T \cup \{G_i\}) - \text{val}(T)), \tag{10}$$

where the summation index $T$ runs over the groups in the sets of groups $\boldsymbol{G}\backslash\{G_i\}$, and $|T|_g$ is the number of groups in $T$. This means that this group Shapley value is simply the game theoretic Shapley value framework applied to groups of features instead of individual covariates, the group Shapley values possess all the Shapley value properties. The practical advantage of working with groups is that the computation of Eq. 10 has a relative computational cost reduction of $2^{d-g}$.

This definition poses the practical question of defining the groups. Here, we group covariates based on a combination of a clustering algorithm and of the dependence measure described in Sect. 3.4. This approach does not however ensure that an effect of dependence among the covariates is totally removed, which may influence the SHAP results as extensively investigated by Aass et al. (2021). To account for it, we rely on the improved kernel SHAP method proposed by Redelmeier et al. (2020) using conditional inference trees (Hothorn et al., 2006) to model the dependence structure of the covariates.

### 3.4 HSIC dependence measure

The number of covariates is 15 (Sect. 2), which is sufficiently large to pose some difficulties regarding the computational time cost of the SHAP approach. As explained in Sect. 3.3, an analysis of the dependence can alleviate this problem by focusing on two levels: (L1) analysis of the pairwise dependence between the covariates to define groups; (L2) analysis of the dependence between the variable of interest and the covariates to filter out covariates of negligible influence (screening analysis). To do so, we rely on the HSIC (Hilbert–Schmidt independence criterion) measure, which can capture arbitrary dependence between two random variables (potentially of mixed type, continuous or categorical). In the following, we describe the main aspects and the interested readers can refer to Gretton et al. (2005) and da Veiga (2015).

Let us associate $X_i$ with a universal reproducing kernel Hilbert–Schmidt (RKHS) space defined by the characteristic kernel function $k_i(.,.)$. The same transformation is associated with $Y$ by considering a RKHS space with kernel $k(.,.)$. We define the HSIC measure as follows:

$$HSIC(X_i, Y) = \text{E}\big(k_i(X_i, X_i')k(Y, Y')\big) + \text{E}\big(k_i(X_i, X_i')\big)\text{E}\big(k(Y, Y')\big)$$
$$-2\text{E}(\text{E}(k_i(X_i, X_i')|X_i)\text{E}(k_i(Y, Y')|Y)), \tag{11}$$

where $(X_i', Y')$ is an independent and identically distributed copy of $(X_i, Y)$, and E(.) is the expectation operator.

The role of the characteristic kernel is here central because it can be defined depending on the type of the considered variables. For continuous variables, the Gaussian kernel is used, and is defined as $\exp(-\lambda\|\mathbf{x} - \mathbf{x}'\|^2)$, with $\lambda$ being the bandwidth parameter chosen as the inverse of the empirical variance of the considered variable. For categorical variables, the identity function is used as a characteristic kernel.



Considering the afore-described level L1, the pairwise dependence is measured by $HSIC(X_i, X_j)_{i \neq j}$. To ease the interpretability, the scaled version of Eq. (11) between 0 and 1 is preferably used, namely the ratio between $HSIC(X_i, X_j)$, and the square root of a normalizing constant equal to $HSIC(X_i, X_i) \cdot HSIC(X_j, X_j)$ as proposed by da Veiga (2015). From the scaled $HSIC(X_i, X_j)_{i \neq j}$, we defined the similarity $S(X_i, X_j)_{i \neq j} = 1 - HSIC(X_i, X_j)_{i \neq j}$ and use the matrix of pairwise similarities as inputs of a clustering algorithm (see Hastie et al., 2009: chapter 14) to define the groups of dependent covariates; for instance

using the Partitioning Around Medoids clustering algorithm denoted PAM (Rdusseeun & Kaufman 1987).

Considering level L2, we rely on the interpretation of $HSIC(X_i, Y)$ from a sensitivity analysis perspective (da Veiga, 2015), namely its nullity indicates that $X_i$ does not influence $Y$. To identify the significant $X_i$, the null hypothesis $'H_0: HSIC(X_i, Y) = 0'$ (against the hypothesis $'H_1: HSIC(X_i, Y) > 0'$) is tested, and the corresponding p-value can be evaluated (El Amri and Marrel, 2021). When the p-value is below a given significance threshold (typically of 5%), it indicates that the null hypothesis

should be rejected, i.e., the considered covariate $X_i$ has a significant influence on the variable of interest $Y$.

## 4 Results

In this section, we apply the proposed procedure (Sect. 3.1) to the two application cases, namely the synthetic (Sect. 4.1) and the real test case of Toulouse city (Sect. 4.2).

### 4.1 Application to the synthetic test case

By construction of the synthetic test case (using Eq. (1)), only the four first covariates described in Table 1 have a significant influence on the synthetic variable of interest. The application of *Step 1* based on the screening analysis using *HSIC* measures (estimated using V-statistics assuming a Gaussian kernel for all variables) confirms this result: the p-value (computed based on the sequential random permutation-based algorithm developed by El Amri and Marrel (2021) with a number of random replicates up to 5,000) of the four first covariates is below 0.1%, whereas the one of the two last ones is respectively 10.3 and

15.6%, i.e. above the significant threshold of 5% (recall that if the p-value<5%, the considered covariate has a significant influence on the variable of interest). Thus, this shows that the number of covariates can be reduced from 6 to 4.

With the restricted number covariates, we train a RF model. As described in Sect. 3.2, we use the conditional mean as the best estimate of the prediction and the inter-quartile width *IQW* for the uncertainty estimate with the 25th and 75th quantile computed using a qRF model. To select the RF parameters, *ns* and $m_{try}$, we repeat 25 times a 10-fold cross validation exercise (Hastie et

al., 2009: chapter 7) by varying *ns* from 5 to 10, and $m_{try}$ from 1 to 4. The number of random trees is fixed at 1,000 (preliminary tests showed that this latter parameter has little influence provided that it is large enough). This tuning procedure selects the pairs (*ns*, $m_{try}$) for which the average relative absolute error is minimised, giving (*ns*=5, $m_{try}$=3) as the combination that results in the lowest error of 6.8% (averaged over 25 replicates of the 10-fold cross-validation) with a frequency of 72% (i.e. 18 replicates out of 25).





Figure 7 shows the best estimate of the true value of the synthetic variable of interest (panel a), the prediction best estimate
(panel b) together with the uncertainty measure (panel c) at 10,000 grid points over the European study area (with a spatial
resolution of ≈13.5 x 13.5 km²). Overall, the RF prediction reproduces relatively well the true spatial distribution (compare
panel a and b in Fig. 7), with an average relative absolute error of about 5%. The uncertainty indicator reaches the highest
values (highlighted by yellow colour in Fig. 7c) where observations are sparsely distributed; in particular in the Alps (zone $Z_1$)

and in the North of Germany (zone $Z_2$). It is important to note that our objective here goes beyond improving the predictive
capability of the RF model: given this level of prediction uncertainty (Fig. 7c), we aim, in the following, to investigate what
are the main drivers of this uncertainty and whether they differ from the ones driving the best estimate of the prediction (Fig.
7b).

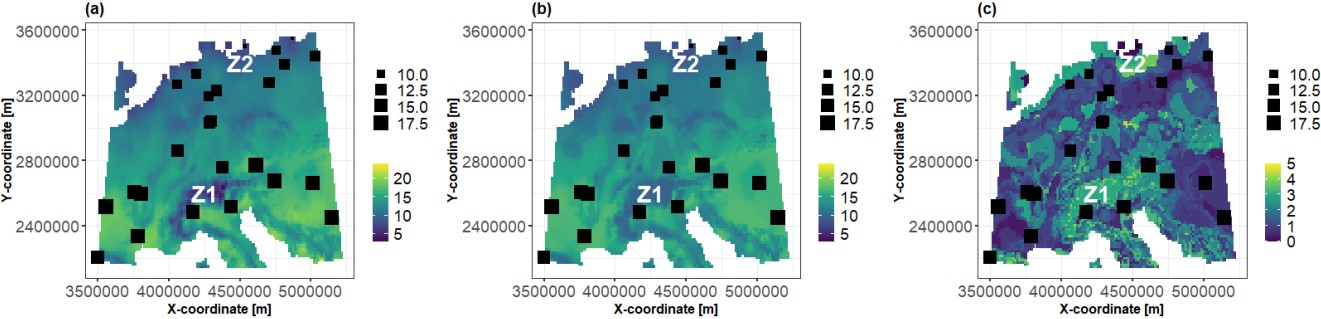

**Figure 7: (a) Spatial distribution of the synthetic variable of interest; (b) Prediction best estimate using the conditional mean of the RF model. (c) Inter-quartile width *IQW* computed using the 25th and 75th quantiles of the qRF model. The 50 soil samples are indicated by the squares whose sizes are proportional to the value of the synthetic variable of interest. The results are more specifically discussed in Sect. 4.1 at the zones indicated by $Z_{1-2}$.**

To further decrease the number of covariates, we analyse the pairwise dependence between the different covariates using the
pairwise *HSIC* dependence measure (assuming with a Gaussian kernel for all covariates). Figure 8 provides the matrix of the
pairwise *HSIC* measures. This shows that some grouping of covariates makes sense, in particular, a group including the
temperature variables, $T_{max}$, and $T_{mean}$, which was expected due to the similarity of both variables. We further justify this
grouping by analysing the average silhouette width of the PAM clustering algorithm. This shows an average silhouette width

of 0.32 and 0.41 considering two and three groups respectively, hence justifying the use of three groups, namely $T_{mean-max}$, and
$T_{range}$, and $P_{wettest}$.





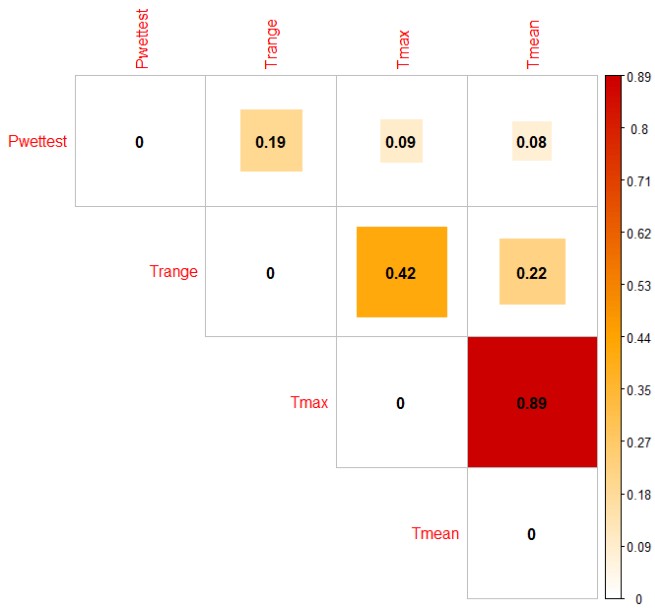

**Figure 8: Matrix of pairwise *HSIC* dependence measure considering the covariates retained after applying the screening analysis for the synthetic test case. The warmer the colour, the higher the dependence.**


Using the trained RF model and the identified groups of covariates, we apply the group-based SHAP approach to carry out, at the 10,000 grid points of the study area, the decomposition (as exemplified at a certain grid point in Fig. 1). To ease the comparison across the study area, we preferably plot the scaled Shapley absolute values (as defined in Sect. 3.3), and use them to map the contribution to the prediction best estimate (i.e. the conditional mean, Fig. 9a) and to the corresponding uncertainty

(i.e. *IQW*, Fig. 9b).

With regard to the prediction best estimate, the upper panels of Fig. 9 show that the three covariate groups have absolute value of the Shapley values in the range [20, 40%] over ≈75% of the whole study area. Figure 9 provides a good illustration that the major contributors to the prediction best estimate and to the uncertainty differ. This is exemplified by the two zones where the uncertainty is the highest (see Fig. 7c). In the central zone around the Alps (zone $Z_1$ in Fig. 9), the major contributor to the best

estimate and to the uncertainty is respectively $P_{\text{wettest}}$ (with contributions of the order of 60%) and the group $T_{\text{mean-max}}$ (with contributions in average of 60-70%). On the other hand, in the North of Germany (zone $Z_2$ in Fig. 9), it is $P_{\text{wettest}}$ that contribute the most to both the best estimate and to the uncertainty. This means that in these two zones, the decisions regarding the characterisation of the covariates are different; this is discussed in more details in Sect. 5.1.



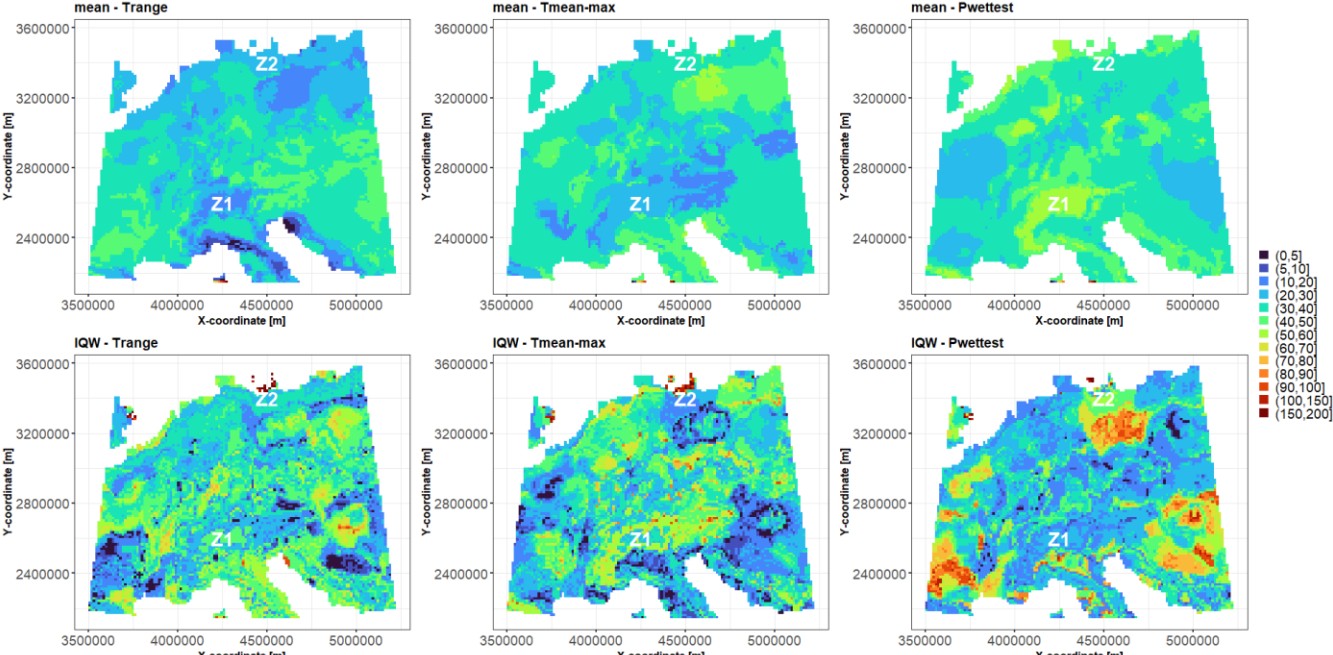

**Figure 9: Scaled Shapley absolute value (in %) for the synthetic test case considering the prediction best estimate using the RF conditional mean (top), and the prediction uncertainty using the qRF inter-quartile width *IQW* (bottom). The results are more specifically discussed in Sect. 4.1 at the zones indicated by $Z_{1-2}$.**

## 4.2 Application to the real case

The application of *Step 1* to the Toulouse real case allows to identify the covariates that have negligible importance regarding
the pollutant concentration. We assume a Gaussian and a categorical kernel for the continuous and the categorical variable
respectively. Figure 6 compares the p-values (using the same computation procedure as for the synthetic case) assuming a
significant threshold of 5%. This shows that the number of covariates can be reduced from 15 to 9 (reduction by 40%).
Several observations can be made:

- The distance to (potentially) polluted sites, $D_{basos}$, has a minor influence contrary to the distance $D_{basias}$ to industrial
  sites (abandoned or in activity). This is due to the high dependence of $D_{basos}$ (whose *HSIC* measure of the order of
  0.93-0.95, Appendix B) with the elevation and the geographical coordinates (Fig. 3 suggests that polluted sites tend
  to locate in relatively low-lying areas in the vicinity of the city centre). In other words, this means that its inclusion
  in the analysis is redundant with respect to the information brought by the covariates to which it is dependent;
- The land use has a strong impact, whereas lithology appears to have little impact (with a p-value larger than 20%, i.e.
  larger than the significance threshold). This is interpreted as being related to the hydrocarbon nature of the pollution,
  which is less related to geological processes contrary to heavy metal pollution for instance;





-   The distance to roads has not been included even though its relation to hydrocarbon concentration was expected. It is due less to its dependence with the other covariates (whose *HSIC* measure is up to 0.14, Appendix B), than to its very dense spatial distribution: the value of this covariate varies very little over a large area (as indicated by the almost
homogeneous colour in Fig. 3), i.e. very few zones are discriminated by this covariate in this case.

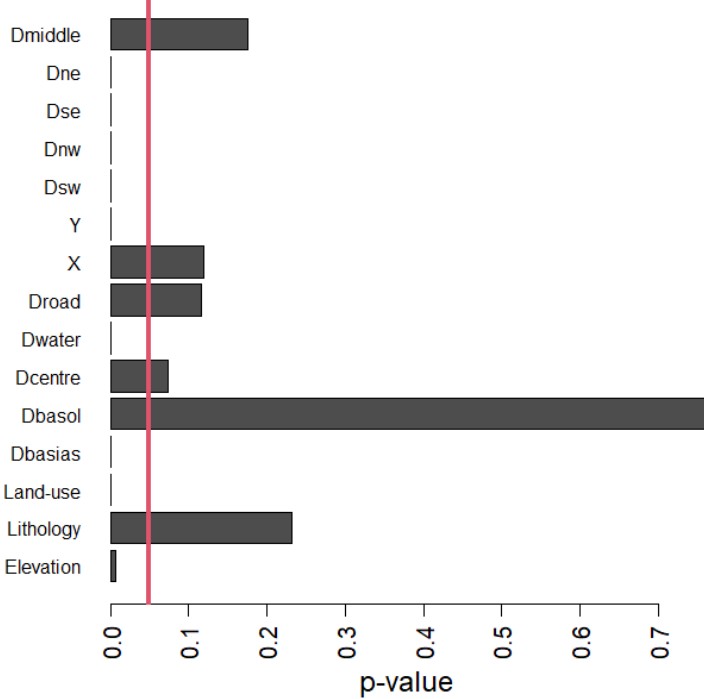

**Figure 10: Screening analysis showing the p-value of the *HSIC*-based test of independence for the Toulouse case. The vertical red line indicates the significance threshold at 5%. When the p-value is below 5%, it indicates that the null hypothesis should be rejected, i.e., the considered covariate has a significant influence on the hydrocarbon concentration.**


With the restricted number covariates (from 15 to 9), we train a RF model. As for the synthetic test case, we use the conditional mean as the best estimate of the prediction and the inter-quartile width *IQW* for the uncertainty estimate with the 25th and 75th quantile computed using a qRF model. One additional difficulty is related to the clustering of the observations as shown in Fig. 3. To alleviate this problem, we follow an approach similar to Bel et al. (2009). First, we use an inverse sampling-intensity

weighting to give more weight to observations in sparsely sampled zones and less weight to observations in densely sampled zones. Second, to estimate the sampling intensity, we use a two-dimensional normal kernel density estimation with bandwidth values estimated based on the rule of thumb of Venables and Ripley (2002). Finally, the inverse sampling-intensity weights (with prior normalisation between 0 and 1) are then used for the RF training during the bootstrap sampling (see Sect. 3.2) to create individual trees with different probability weights (by following a method similar to Xu et al., 2016).



To select the RF parameters, *ns* and *m*<sub>try,</sub> we use the same cross validation exercise as for the synthetic case. This tuning procedure selects the pairs (*ns*, *m*<sub>try</sub>) for which the average relative absolute error is minimised, giving (*ns*=5, *m*<sub>try</sub>=2) as the combination that results in the lowest error of ≈12% (averaged over 25 replicates of the 10-fold cross-validation) with a frequency of 76% (i.e. 19 replicates out of 25). Figure 11 shows the prediction (panel a) of the hydrocarbon concentration together with the uncertainty measure (panel b) at the grid points over the city (with a spatial resolution of 100 x 100 m²).

We note that a large proportion of the city has a predicted concentration varying between 1 to 2 (on log₁₀ scale) at the exception of the south-eastern part where the concentration is predicted with values >3. In this zone, the uncertainty of the prediction is the highest with values ranging from up to ≈2.5. Outside this zone, a large proportion of the study area has uncertainty estimates <1.0 with some zones with uncertainty <0.01 (in particular in the vicinity of the observations).

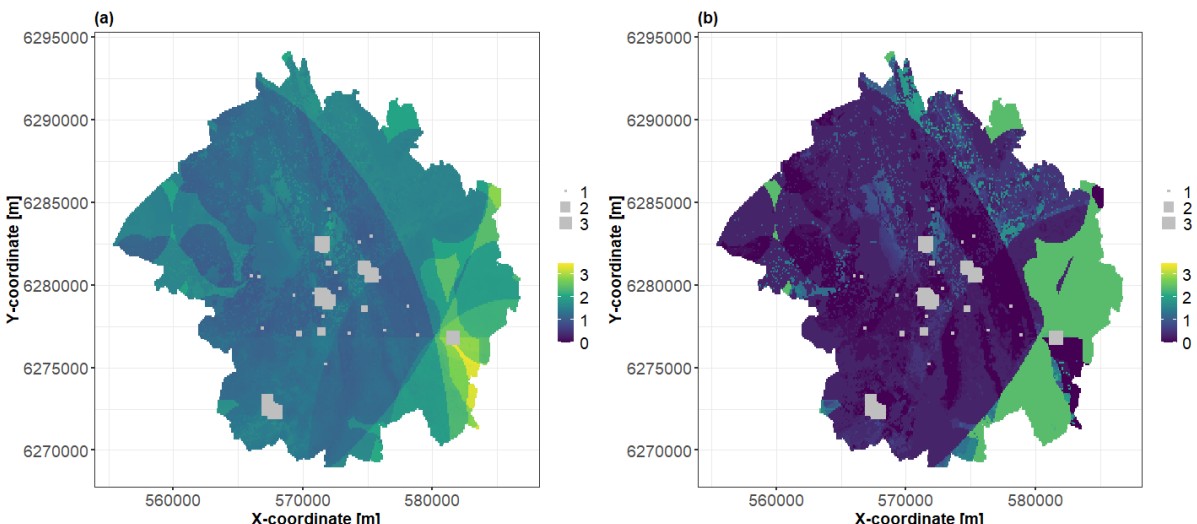


**Figure 11: (a) Prediction best estimate of the hydrocarbon concentration (log₁₀ scale) for Toulouse city using the conditional mean of the RF model. The soil samples are indicated by the squares whose sizes are proportional to the log₁₀ of the C10-C4 hydrocarbon concentration; (b) Inter-quartile width *IQW* computed using the 25th and 75th quantiles of the qRF model (expressed on the same scale as the hydrocarbon concentration).**


To further decrease the number of covariates, we analyse the pairwise *HSIC* dependence measure (assuming a Gaussian and a categorical kernel when the considered variable is continuous or categorical variable). Figure 12 provides the matrix of pairwise *HSIC* measures, and shows that some grouping of covariates makes sense, in particular, a group including the distance variables, *D*<sub>ne</sub>, *D*<sub>se</sub>, *D*<sub>nw</sub>, *D*<sub>sw</sub> (introduced to better account for the spatial dependence, see Sect. 2), with the *Y*-coordinate and

the elevation (they all present moderate-to-high *HSIC* values>0.50). We also note that the land-use has low dependence with the other variables (low *HSIC* value <0.10). We further justify this grouping by applying the PAM clustering algorithm with





a number of groups selected to maximise the average silhouette width (Appendix B). This shows that the average silhouette width is maximized when considering 3 groups, namely:

- The group named *Elv-D-Y* which includes $D_{ne}$, $D_{se}$, $D_{nw}$, $D_{sw}$, the *Y*-coordinate and the elevation. This result is 400 consistent with the overall spatial distribution of the elevation which mostly increase from west to east (Sect. 2.2);
- The group named $D_{basias-water}$ which includes $D_{basias}$ and $D_{water}$. This result can be intuitively understood as being linked to the general tendency of industrial sites to locate close to a water supply;
- The l*and-use* covariate.

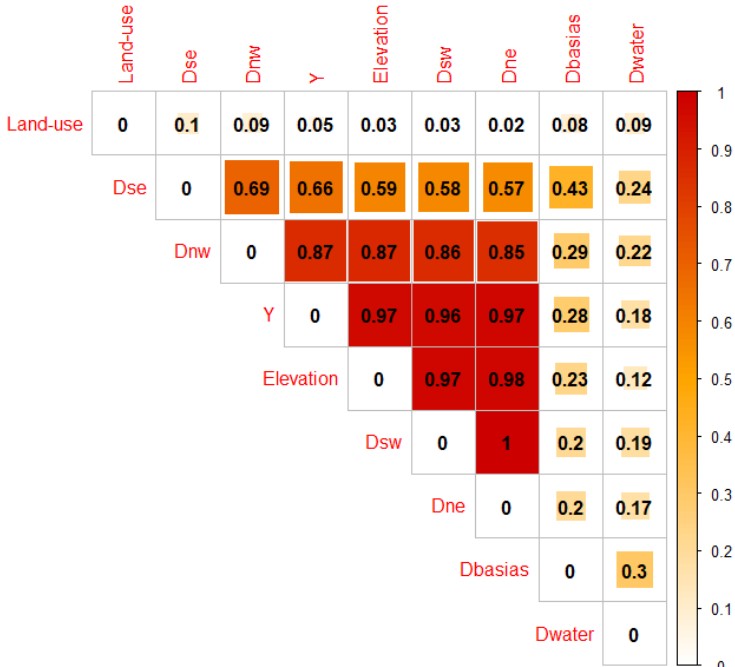

**Figure 12: Matrix of pairwise *HSIC* dependence measure considering the covariates retained after applying the screening analysis for the Toulouse test case. The warmer the colour, the higher the dependence.**

Using the trained RF model and the identified groups of covariates, we apply the group-based SHAP approach to carry out, at the >45,000 grid points of the study area, the decomposition (as exemplified at a certain grid point in Fig. 1). As for the 410 synthetic test case, we compute the scaled Shapley absolute values, and use them to map the contribution to the prediction best estimate (i.e. the conditional mean, Fig. 13a) and to the corresponding uncertainty (i.e. *IQW*, Fig. 13b). The areas with very low *IQW* values have very high scaled Shapley absolute values (by definition), and to ease the analysis, we exclude these zones with *IQW*<0.01 outlined in dark red in Fig. 13.





With regard to the prediction best estimate, the upper panels of Fig. 13 show that the three covariate groups contribute, to some
extent, with proportions of more or less the same order of magnitude; the absolute value of the Shapley values being in the
range [25, 50%] over >50, >67 and >60% of the whole study area for the *Elv-D-Y*, *Land-use* and $D_{\text{basias-water}}$ covariate groups,
respectively. This is also confirmed by the analysis of the boxplots in Appendix B. The prediction uncertainty in the eastern
part of the city, i.e. in the zone indicated by $Z_1$ in Fig. 13 (where the predicted concentration reach the highest) and in the zone
indicated by $Z_2$, is mainly explained by *Elv-D-Y* with a contribution of the order of 50-75% (i.e. in a zone where the elevation
is particularly high, see Fig. 4). The western part of the city (indicated by $Z_3$) is also particularly outlined with a large
contribution of the $D_{\text{basias-water}}$ group (and minor influence of *Elv-D-Y*), i.e. in a zone which is particularly farthest from industrial
sites or rivers (Fig. 4).

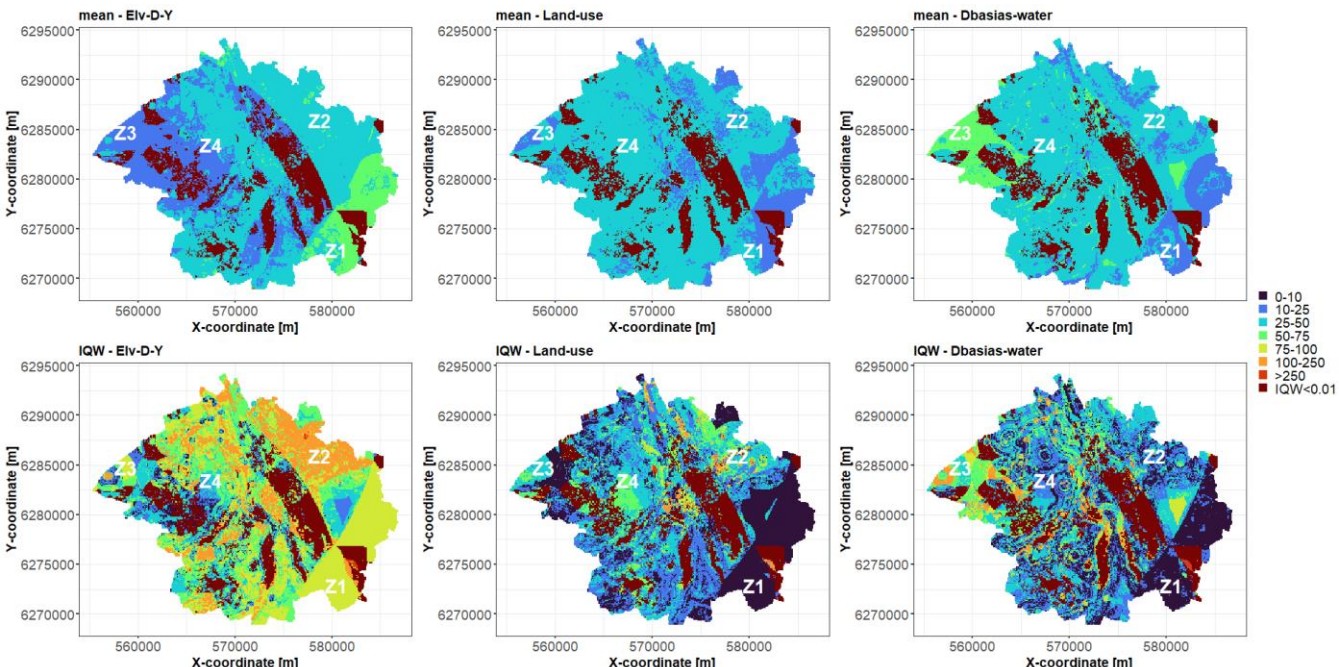

**Figure 13: Scaled SHAP absolute value (in %) for the Toulouse test case considering the prediction best estimate using the RF**
**conditional mean (top), and the prediction uncertainty using the qRF inter-quartile width *IQW* (bottom). The areas in dark red**
**correspond to the grid points where *IQW* is low, <0.01. The results are more specifically discussed in Sect. 4.2 at the zones indicated**
**by $Z_{1-4}$.**

Regarding the prediction uncertainty, the spatial patterns of the scaled Shapley values are more complex. Overall, the
prediction uncertainty appears to be mainly impacted by the *Elv-D-Y* group with ~45% of the whole area having a scaled
Shapley absolute value >75% (as indicated by the yellow / orange colour in the bottom, left panel of Fig. 13) including the
zone of the largest predicted concentration values (zone $Z_1$). The western part of the city (zone $Z_3$) is also particularly outlined,





namely, where the *Elv-D-Y* group has less impact (as indicated by the dark blue colour), i.e. in a relatively low-lying area. In
this zone, it is the $D_{\text{basias-water}}$ group that explains the most the prediction uncertainty with scaled Shapley absolute values >75%.
The scaled Shapley absolute values can even be very high, above 100% (outlined by the orange/red colour in Fig. 13): this is
an indication that the other covariates' group have negative scaled Shapley values (named "negative contributors") so that the
scaled Shapley values of all covariates' groups sum to 100% (i.e. the unscaled Shapley values sum to the total prediction
uncertainty, see Sect. 3.3). This is analysed in Fig. 14, which shows that almost 50% of the study area have negative
contributors, which mainly correspond to land-use with scaled Shapley value down to -50% mostly in the northern part.
Finally, we note that at a finer spatial scale, the contributions of the groups are highly variable, as indicated by the circular
spatial patterns in the bottom, middle and right panels of Fig. 13, which appears in relation to the fine-scale spatial distribution
of $D_{\text{basias}}$ and $D_{\text{water}}$ (Fig. 4).

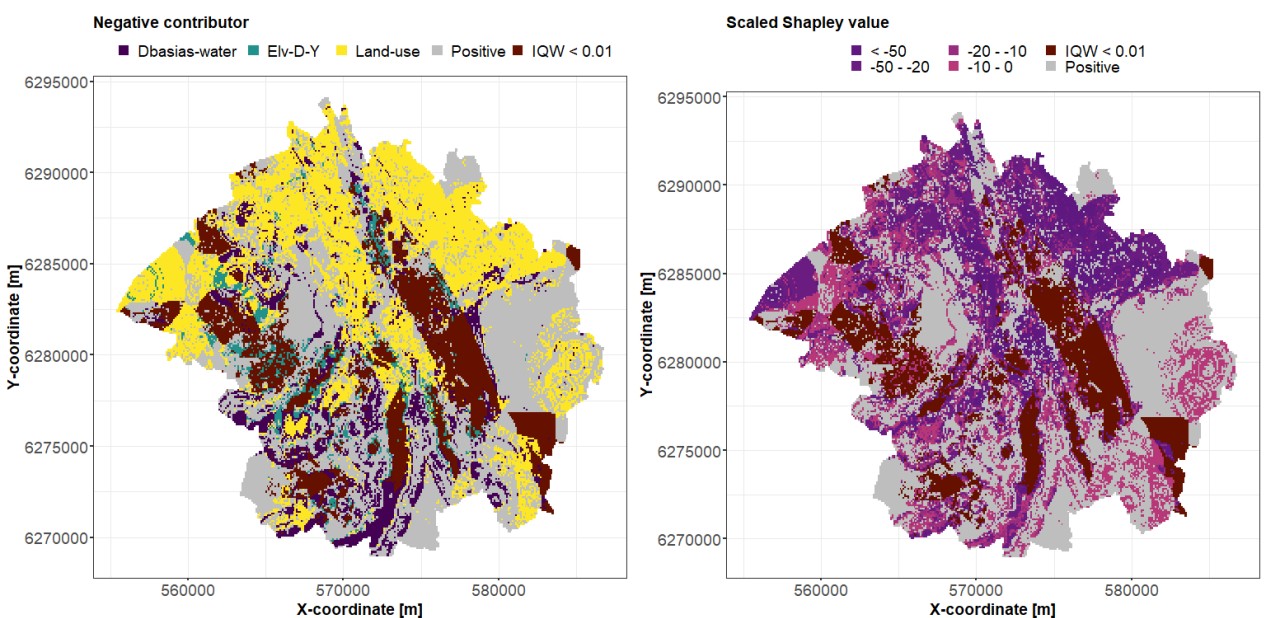


**Figure 14: Identification of the group of covariates (left panel) with negative Shapley values (right panel) for the prediction
uncertainty at Toulouse city. The areas in dark red correspond to the grid points where *IQW is* low, <0.01. The areas in grey
correspond to grid points with positive Shapley values.**

## 5 Discussion

In this section, we discuss in more details the usefulness of the results described in Sect. 4 to support decision making regarding
uncertainty management (Sect. 5.1) as well as the added value of the proposed grouping-based procedure (Sect. 5.2).



## 5.1 Usefulness of the results

To date, the Shapley values have been used to explain individual predictions related to a certain instance of the covariates by computing the contribution of each of them to the prediction. Translated for DSM, the Shapley values can be used to answer why the spatial ML model came to a certain value of the soil or chemical properties at a certain spatial location. As discussed by Wadoux and Molnar (2022), the use of Shapley values has the potential to constitute a key ingredient in the toolbox of environmental soil scientists to improve the interpretation of ML-based DSMs; in particular by providing insights into the underlying physical processes that drive soil variation.

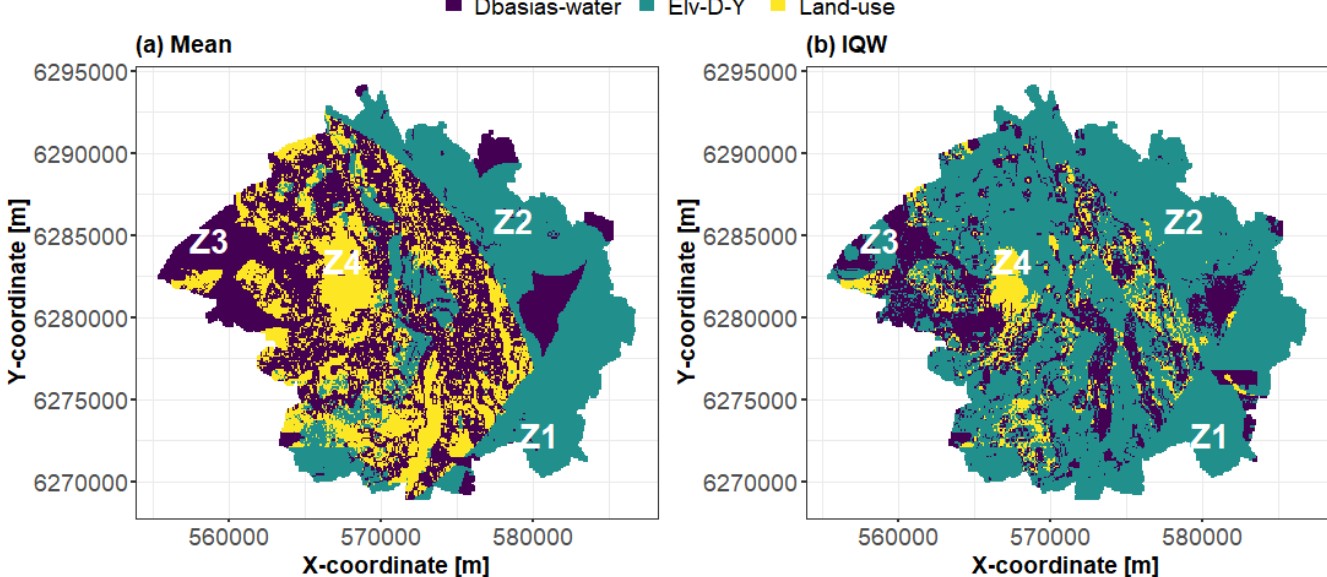

**Figure 15: Spatial distribution of the most important group of covariates with respect to the scaled Shapley absolute values for the prediction best estimate (mean, panel a) and for the prediction uncertainty (*IQW*, panel b) at Toulouse city. The results are more specifically discussed in Sect. 4.2 at the zones indicated by *Z*$_{1-4}$.**

In this study, we complement this type of analysis by addressing the "why" question with respect to the prediction uncertainty, i.e. by explaining why the spatial ML model is confident. For this goal, SHAP approach to estimate the Shapley values is applied to decompose the uncertainty indicator provided by the ML spatial model. The attribution results are expected to facilitate the communication between environmental soil scientists and stakeholders, which is a key in the inclusion of these new digital soil map products in current practices (see e.g., discussion by Arrouays et al., 2020). The SHAP results are expected to improve the framing of the prediction results together with the associated uncertainty. These narratives can follow the example of the synthetic test case (Fig. 1): "The predicted value of 12.20 is mainly attributable (by a positive factor of almost 50%) to the maximum and mean temperature of the warmest quarter of the year. The confidence in this result (measured by



the uncertainty indicator of 1.98) is explained by the diurnal range (by almost 50%). To further increase this confidence (and decrease the uncertainty), next efforts should concentrate on the characterisation of this particular covariate".

The second implication of our study is in terms of uncertainty management. Our application to the hydrocarbon concentration mapping at Toulouse, as well to the synthetic test case, reveals that the determining contributors to either situation (prediction best estimate or uncertainty) may not be necessarily the same. This is shown in Fig. 15, which provides the maps of the most important group of covariates (with respect to the scaled Shapley absolute values) for the real case. This shows that the prediction uncertainty is dominated by *Elv-D-Y* over almost 70% of the entire study area, whereas the best estimate is influenced by this group of covariates over less than 35%. This means that different assumptions along the data cycle of the urban soil geochemical background (see e.g., the discussion by Belbeze et al., 2023: Sect. 5) have different practical implications. An illustrative example is the central part (zone $Z_4$ in Fig. 13), where land-use and $D_{basias-water}$ jointly influence the prediction best estimate, but it is only the land-use that has the most significant influence (>50%) on the uncertainty (outlined by the green-coloured patch in the bottom, central panel of Fig. 13). This means that, to increase the prediction confidence, the environmental soil scientists should concentrate the effort on the phase of data interpretation, and more specifically on the processing of land use datasets; for example by using a finer grouping of the land-use categories, see Sect. 2. On the other hand, the environmental soil scientists should concentrate the effort on two aspects regarding the prediction values, namely data interpretation (as for uncertainty), and data collection (sampling), because in this zone, only few observations are available in the training dataset (only 10 observations have $D_{basias}$ and $D_{water}$ above 500 and 1,400m respectively). This dichotomy between drivers of best estimate and uncertainty is also illustrated by the synthetic test case in the central zone around the Alps (zone named by $Z_1$ in Fig. 9).

Conversely, in the zones where the contributors to either situation are the same, the environmental soil scientists can be confident that any future decisions regarding a phase of the data cycle (such as data collection, interpretation, etc.) should affect similarly the best estimate and the confidence in this prediction. This is illustrated by the *Elv-D-Y* group (identified as the main contributor to either situation, Fig. 13, top; in particular in zones $Z_1$ and $Z_3$): increasing the accuracy of the digital elevation model or adopting alternative approaches to incorporate spatial dependence (Behrens et al., 2018) should in this case jointly benefit either situation. This is also illustrated with the synthetic test case, where $P_{wettest}$ (Fig. 9, right) has the largest contribution to either situation in the North of Germany (zone $Z_2$).

Finally, the identification of the zones where groups of covariates have negative contributions (i.e. negative contributors as outlined in Fig. 14 for the real case) is worth conducting because those directly participate to the decrease of the prediction uncertainty.

## 5.2 Added value of grouping

One major drawback of the SHAP approach (regardless of the target, prediction best estimate or uncertainty) is its computational load that is directly related to the number of covariates (e.g. Jullum et al., 2021). In the real case, the SHAP computational complexity is proportional to $2^{15}=32,768$. The application of *Step 1* (screening analysis; Fig. 10) allows a





decrease of the number of features from 15 to 9, hence a relative computational cost reduction of $2^{15-9}$=64. The grouping based

on the dependence analysis (*Step 3* of the proposed procedure) then implies an additional relative computational cost reduction

of $2^{9-3}$=64. This has clear practical benefits in the Toulouse case, where the SHAP decomposition has to be applied at >45,000

spatial locations. To illustrate the gain in computational time cost (CPU time), we run SHAP for the nine important covariates

(without grouping) at 100 randomly selected grid points (on a Windows Desktop x-64 with PC – Intel® Core™i5-13600H,

2,800 MHz, 12 Core(s), 16 Logical Processor(s) with 32GB physical RAM) which led to an average CPU time of 2.15 seconds.

Extrapolated to the whole set of grid points, this means that the analysis would have required more than 1 day of calculation

to be compared to ≈0.5 hour for the group-based SHAP (with an average CPU time of 0.046 seconds). Applications to national-

scale studies like the one by Wadoux et al. (2023) for France (with several hundreds of thousands of grid points) or to global-

scale maps of the environment (like the ones provided by Poggio et al., 2022) should in this situation be facilitated by the

group-based SHAP approach. Two implications of the grouping are worth underlying: (1) it avoids the use of sampling-based

algorithms that introduce an approximation error of the Shapley values; (2) it limits the use of high performance computing

architecture, which should be thought in terms of energy consumption, which is becoming a growing concern for scientific

computing (see discussion by Jay et al., 2024).

However, one major limitation of the grouping is the loss in interpretability of the SHAP results, because the separate

contribution of each member of a given group is more difficult to understand. This is exemplified with the *Elv-D-Y* group

whose influence should be interpreted by analysing the joint evolution of six covariates. For group with less members, some

general behaviour can however be identified, like the $D_{\text{basias-water}}$ group, whose influence can be analysed with respect to the

general tendency of industrial sites to locate close to a water supply. The same applies for the group $T_{\text{mean-max}}$ for the synthetic

test case. This lack of interpretability is even worsened by the additional difficulty to perform a "partial dependence" analysis

using the group-based SHAP (i.e. a scatterplot of the average of the Shapley values against the value of a single covariate to

model the functional form of the association, see examples provided by Wadoux et al., 2023: Figure 3).

Remedies to this problem can focus on the second grouping option by Jullum et al. (2021) based on underlying

knowledge/expertise (i.e. grouping covariates that make sense with respect to the problem at hand). In the synthetic test case,

the grouping of $T_{\text{mean}}$ and $T_{\text{max}}$ could have be done beforehand without resorting to the pairwise dependence analysis. On the

other hand, when this knowledge/expertise is lacking, an alternative can rely on a two-stage analysis: (1) the first stage focuses

on the group-based SHAP approach, and highlights certain areas of interest; (2) the second stage applies SHAP to all covariates

in these areas. This is illustrated with Fig. 16 in the zone where $IQW$ is the highest in the south-eastern part (zone $Z_3$ in Fig.

11b). On the one hand, Fig. 16a provides the group-based SHAP analysis and outlines the major role of the *Elv-D-Y* group.

On the other hand, Fig. 16b allows identifying that the distances to the south- and north-western corner ($D_{\text{sw}}$ and $D_{\text{nw}}$ included

in the *Elv-D-Y* group) are the major drivers of uncertainty. In the studied area, $D_{\text{sw}}$ and $D_{\text{nw}}$ take very large values, i.e. over a

range of values that are sparsely represented in the training database: the soil samples (Fig. 3) are preferably located in the

vicinity of the city centre. The moderate influence of $D_{\text{basias-water}}$ is here explained by a low-to-moderate importance of $D_{\text{basias}}$

and of $D_{\text{water}}$. Finally, we note that both analyses agree on the low importance of land use. It is however only because of the




application of the SHAP approach at the second stage that the elevation is identified here as a low contributor (despite belonging to the very influential *Elv-D-Y* group). This is explained by the high elevation value, which is far higher than those observed in the training dataset.

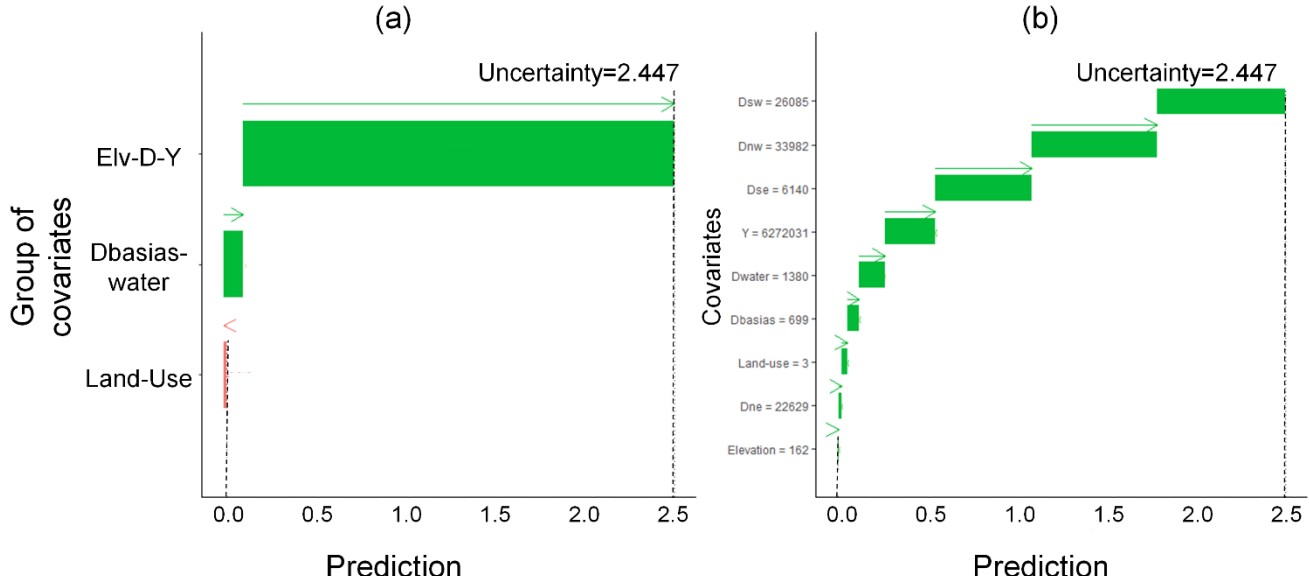

**Figure 16: SHAP-based decomposition of the prediction uncertainty measured by the inter-quartile width *IQW* (modelled by the qRF model trained in Sect. 4.1) for the pollutant concentration (log$_{10}$-scale) in zone $Z_3$ of Toulouse city where *IQW* is predicted as the highest (of 2.447): (a) Application of the group-based SHAP approach; (b) Application of SHAP without grouping. Each horizontal bar provides the contribution to the prediction (indicated by the vertical dotted line) of each covariate or group of covariates (indicated on the vertical axis).**

## 6 Concluding remarks

Providing insights into the uncertainty impacting DSM is a key challenge that require appropriate diagnostic tools. In the effort to complement the toolbox of environmental soil scientists, we have assessed in this study the feasibility of the SHAP approach to quantify the contribution of the covariates to the machine-learning-based prediction uncertainty at any location of the study area. To do so, two main practical difficulties have been addressed by relying on the simple-but-efficient grouping approach of Jullum et al. (2021), namely: (1) the high computational burden of SHAP, and (2) the influence of covariates' dependence. On this basis, we have explored the benefit of analysing the prediction best estimate along the prediction uncertainty using the real case of pollution concentration mapping in the city of Toulouse as well as a synthetic test case. Our results have shown that the drivers of the prediction best estimate are not necessarily the ones that drive the confidence in the predictions: this means that decision in terms of data collection and covariates' characterisation may differ depending on the target, the

prediction best estimate or the confidence/uncertainty and the way in which the results of the prediction (and their uncertainty) are communicated.

Integrating SHAP at a fully operational level needs however to consider several lines of improvement. First, we have focused on a unique uncertainty indicator, here the inter-quartile width, but in some situations, this may not be representative enough of the total uncertainty, and additional developments are necessary to integrate the entire prediction probability distribution within the SHAP setting. The use of information theoretic variant of Shapley values as investigated by Watson et al. (2023) may here be helpful. Second, we have focused on one type of machine learning model: the quantile RF model, but alternative

approaches should be considered in future research works: different types of ML model (for instance deep learning techniques which have shown promising results, see e.g. Kirkwood et al. (2022)), or improved approaches, in particular to deal with complex samples' distribution like clustering (see e.g. de Bruin et al. (2022) and references therein), or different uncertainty measures, for example based on geostatistical methods (using the kriging variance or statistical quantities derived from stochastic simulations, see e.g. Chilès and Delfiner (2012)), or based on Bayesian techniques (see Abdar et al., 2021 for deep

learning techniques) or based on data-driven approaches like cross-validation procedures (Ben Salem et al., 2017). These future studies are made possible thanks to the model-agnostic nature of SHAP.

**Author contributions**

JR designed the concept. SB provided support for the data processing, for the implementation of the machine learning model, and the application to the Toulouse case. JR undertook the statistical analyses. The results were analysed by JR, SB, and DG.

JR wrote the manuscript; SB and GD reviewed the manuscript.

**Competing interests**

The authors declare that they have no conflict of interest.

**Code/Data availability**

Sources of data of the covariates are listed in Table 1 and Table 2. We provide the R scripts to run the synthetic test case in the

form of an R markdown on the Github repository: https://github.com/anrhouses/groupSHAP-uncertainty based on the vignette of the R package *CAST* available at: https://hannameyer.github.io/CAST/articles/cast02-AOA-tutorial.html. The data of the Toulouse test case have however restricted access restriction.



**Acknowledgements**

We acknowledge financial support of the French National Research Agency within the HOUSES project (grant N°ANR-22-CE56-0006). We are grateful to the "Ministere de la Transition Ecologique et Solidaire, Direction Générale de la Prévention des Risques (MTES/DGPR)" and Toulouse Metropole for letting us use the data that supported the study (Belbeze et al., 2019).

**Appendix A Implementation**

The R package *ranger* developed by Wright & Ziegler (2017) is used to train the RF models as well for the predictions, and quantile estimates. The R package *sensitivity* (https://cran.r-project.org/web/packages/sensitivity/index.html) is used to implement the HSIC-based analysis (screening and dependence). The R package *shapr* developed by Sellereite et al. (2023) is used to implement the group-based SHAP approach. The R package *cluster* by Maechler et al. (2023) is used to implement the PAM clustering method. A R markdown based on the vignette of the R package *CAST* (available at: https://hannameyer.github.io/CAST/articles/cast02-AOA-tutorial.html) is provided on the Github repository: https://github.com/anrhouses/groupSHAP-uncertainty.



**595     Appendix B Complementary analysis for the Toulouse real case**

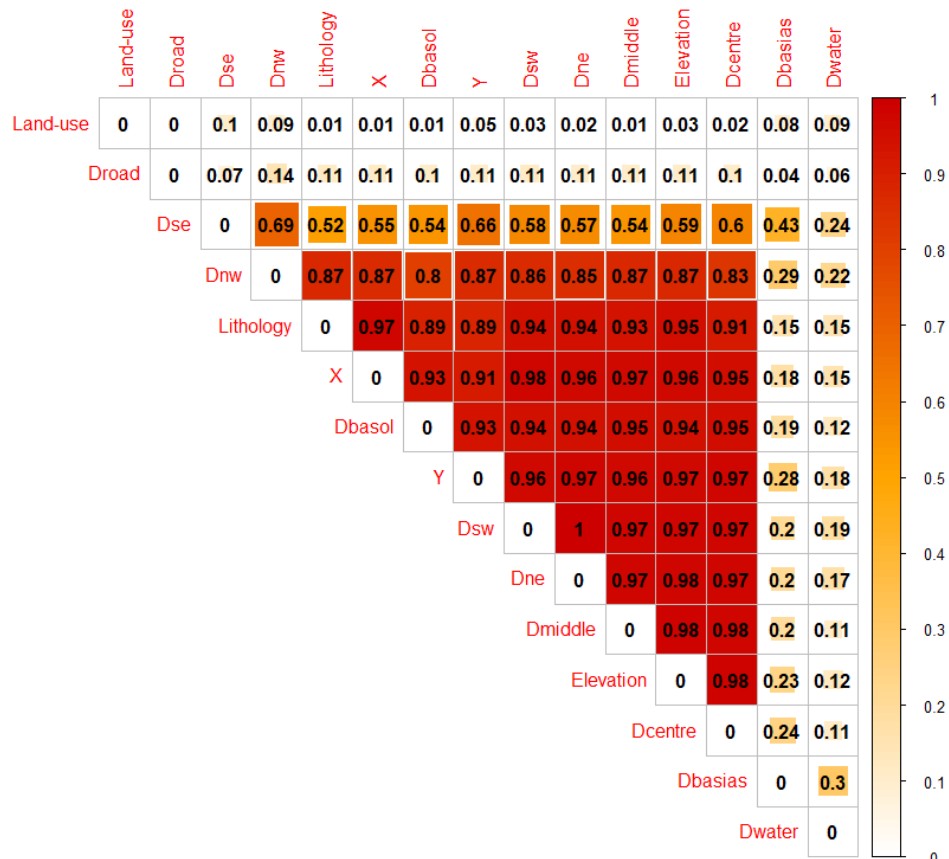

**Figure B.1 Pairwise dependence matrix for all covariates.**



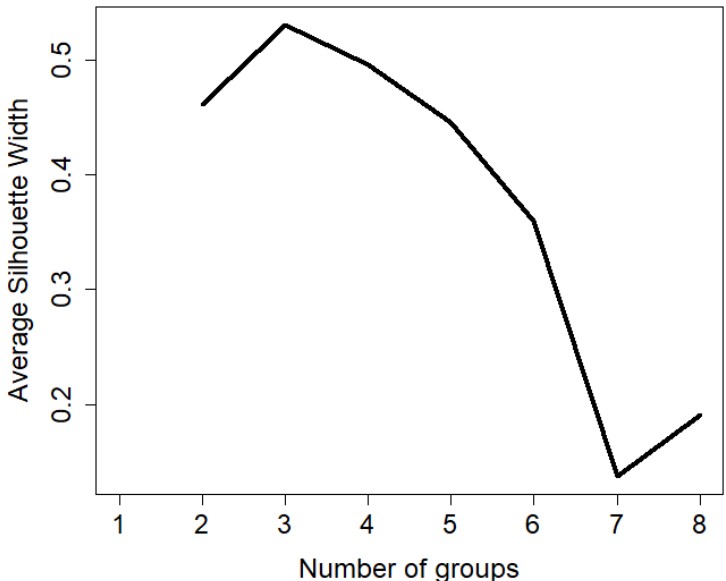

**Figure B.2 Selection of the number of groups.**


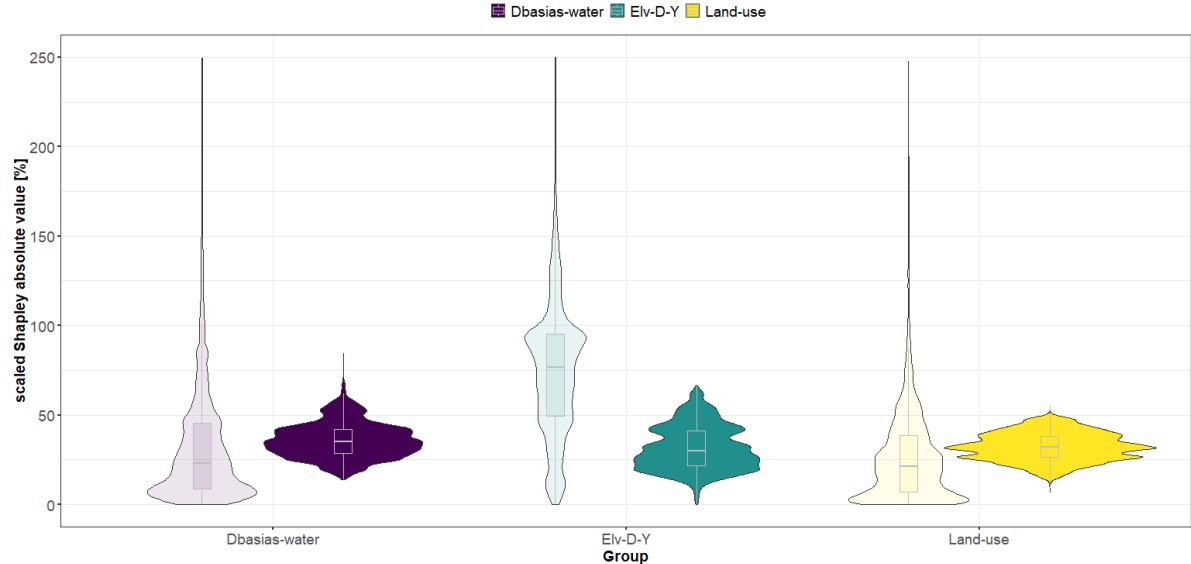

**Figure B.3: Statistics over the study area (>40,000 grid points) of the scaled Shapley absolute value for the three groups of covariates using the RF conditional mean (bold colours), and the prediction uncertainty using the qRF inter-quartile width *IQW* (light colours).**



## Appendix C List of acronyms

- *IQW*: inter-quartile width
- *HSIC*: Hilbert–Schmidt Independence Criterion
- *ML*: Machine Learning
- *qRF*: quantile Random Forest
- *RF*: Random Forest
- *SHAP*: SHapley Additive exPlanation

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
