# Peer review of "Insights into the prediction uncertainty of machine learning-based digital soil mapping through a local attribution approach"

_EGUsphere, 2024_

## Author Response (AR1)

**Replies to Editor's comments on "Insights into the prediction uncertainty of machine-learning-based digital soil mapping through a local attribution approach" (egusphere-2024-323)**

We would like to thank the Editor for giving us the opportunity to correct our manuscript based on the constructive comments of both Referees. We agree with most of the suggestions and, therefore, we have modified the manuscript to take on board their comments.

**Editor:**

*The two referees made useful suggestions for improvement. I agree with many of the comments from R1, in particular the use of variable selection techniques prior to model fitting. It is usually not necessary for tree-based models like QRF, and perhaps even less so for a small number of covariates. What is the added value of the HSIC test for machine learning models? Authors should consider removing this step.*

As rightly stated by the Editor, the problem is not related to the use of RF, because RF models can handle a large number of covariates. The problem is the computational burden for the computation of the Shapley values. This is our motivation for proposing to complement the analysis with the *HSIC*-based screening analysis.

In the manuscript, different elements have been clarified.

(1) In Sect. 3.3 'overall procedure', we have clarified the problem of computational burden.

(2) In Sect. 5.2 'applicability to global scale projects', we underline the following aspects:

- Many real case studies with a moderate number of covariates of the order of 10-20 as in our case, have implemented approaches to select covariates. This can be done using different methods, for instance recursive feature elimination, forward feature selection, and RF importance measures as illustrated by the study by Meyer et al. (2019) and Dornik et al. (2022);

- For large-scale projects with hundreds of covariates (see comment of Referee #1), we now analyse different options on page 22 as follows:

[…] "we run SHAP for the Toulouse test case using the nine important covariates (without grouping) at 100 randomly selected grid points (on a Windows Desktop x-64 with a PC – Intel® Core™ i5-13600H, 2,800 MHz, 12-core, 16 logical processor(s) with 32 GB physical RAM), which led to an average CPU time of 2.15 seconds. Given the constraints of global-scale studies, a direct SHAP analysis would require at least 200 days of calculation on a single laptop. The first solution relies on the use of a high-performance computing architecture, as proposed by Wadoux et al. (2023). A second option involves approximating the Shapley values using, for instance, sampling algorithms (Chen et al., 2023), with some approximation errors opposite to those of the exact method used here. A third option explored in this study is the combination of

screening analysis and a grouping approach. Although RF models can handle a large number of covariates, eliminating the covariates before calculating the Shapley values has a clear benefit for saving CPU time. In the real case, the SHAP computational complexity is proportional to $2^{15}=32,768$. The application of screening analysis (Fig. 8) decreases the number of features from 15 to 9, resulting in a relative computational cost reduction of $2^{15-9}=64$. An additional step of grouping is proposed here, with the primary objective of facilitating interpretation. Interestingly, Wadoux et al. (2023) also presented Shapley values for groups of covariates (mean climate, climate extremes, vegetation, topography, etc.), as indicated in Figure 6 of their study. By grouping before calculating the Shapley values, an additional relative computational cost reduction can be achieved. In the Toulouse case, this implies a cost reduction of $2^{9-4}=32$, and the analysis required less than one hour for the group-based SHAP (with an average CPU time of 0.054 seconds). Given the constraints of global-scale studies, this approach would here require less than 7 days of calculation on a single laptop. With the growing concern regarding energy consumption (see, e.g., Jay et al., 2024) for scientific computing, this option provides soil scientists with efficient, energy-saving analytical tools although it requires a careful identification of the covariates of negligible influence as well as the definition of groups".

(3) In the concluding remarks, we underline as a perspective, on page 23, that "the implementation to global scale projets still remains challenging and deserves further work to find a comprise between accuracy, efficiency and interpretability with particular attention to estimation algorithms (Chen et al., 2023) with a potential combination with screening and grouping analysis."

(4) In Sect. 4.2, we underline that the results of the *HSIC* screening analysis provide valuable information regarding the particularities of our problem of soil pollution.

- It confirms the negligible influence of the lithology in relation to the nature of the hydrocarbon pollution, which is less related to geological processes contrary to heavy metal pollution for instance;

- It indicates that the distance to roads has a minor role. It is due less to its dependence with the other covariates, whose *HSIC* measure is up to 0.14 (Supplementary Materials B), than to its very dense spatial distribution: the value of this covariate varies very little over a large area as indicated by the almost homogeneous colour in Fig. 3, i.e. very few zones are discriminated by this covariate in this case.

(5) In Sect. 3, we clarify that the screening analysis using *HSIC* dependence measure has proven to be very efficient in the machine learning community (see e.g. Gretton et al., 2005) with applications in multiple domains, e.g. atmospheric pollution (Fellmann et al., 2023); environment (Lambert et al., 2024); geochronology (Herrando-Pérez & Saltré 2024); nuclear safety (Marrel and Chabridon, 2021); deep learning and image analysis (Novello et al., 2022); geothermics (Rohmer et al., 2023).

The key advantages in our case are:

- *HSIC* measure can capture arbitrary dependence without resorting to some assumptions such as linearity, monotonicity;

- *HSIC* measure can handle random variables potentially of mixed type, continuous or categorical;

- *HSIC* measure avoids the use of RF importance measures, which show some limits as extensively discussed, among others, by Ishwaran (2007), Strobl et al., (2007), Benard et al. (2022). This aspect has also been clearly underlined by Meyer et al. (2019).

For these different reasons, we believe that proposing a screening analysis prior to the SHAP implementation remains useful for operational implementation and deserves to be included as a step in the procedure, although not compulsory.

**References**

Bénard, C., Da Veiga, S., & Scornet, E. (2022). Mean decrease accuracy for random forests: inconsistency, and a practical solution via the Sobol-MDA. Biometrika, 109(4), 881-900.

Chen, H., Covert, I. C., Lundberg, S. M., and Lee, S. I.: Algorithms to estimate Shapley value feature attributions. Nature Machine Intelligence, 5(6), 590-601, 2023.

Fellmann, N., Pasquier, M., Blanchet-Scalliet, C., Helbert, C., Spagnol, A., & Sinoquet, D. (2023). Sensitivity analysis for sets: application to pollutant concentration maps. arXiv preprint arXiv:2311.16795.

Herrando-Pérez, S., & Saltré, F. (2024). Estimating extinction time using radiocarbon dates. Quaternary Geochronology, 79, 101489.

Ishwaran, H. (2007). Variable importance in binary regression trees and forests. Electronic Journal of Statistics, 1:519–537.

Lambert, G., Helbert, C., & Lauvernet, C. (2024). Quantization-based LHS for dependent inputs: application to sensitivity analysis of environmental models. https://ec-lyon.hal.science/hal-04546338/file/Article_QLHS.pdf

Marrel, A., & Chabridon, V. (2021). Statistical developments for target and conditional sensitivity analysis: application on safety studies for nuclear reactor. Reliability Engineering & System Safety, 214, 107711.

Meyer, H., Reudenbach, C., Wöllauer, S., & Nauss, T. (2019). Importance of spatial predictor variable selection in machine learning applications–Moving from data reproduction to spatial prediction. Ecological Modelling, 411, 108815.

Novello, P., Fel, T., & Vigouroux, D. (2022). Making sense of dependence: Efficient black-box explanations using dependence measure. Advances in Neural Information Processing Systems, 35, 4344-4357.

Rohmer, J., Armandine Les Landes, A., Loschetter, A., & Maragna, C. (2023). Fast prediction of aquifer thermal energy storage: a multicyclic metamodelling procedure. Computational Geosciences, 27(2), 223-243.

Strobl, C., Boulesteix, A.-L., Zeileis, A., & Hothorn, T. (2007). Bias in random forest variable importance measures: illustrations, sources and a solution. BMC Bioinformatics, 8:25.

*The level of English throughout the manuscript is currently not sufficient to allow a clear understanding of the content. You may want to ask a proficient reader to check your manuscript for grammar, style and syntax before resubmitting it. There are also several typos.*

The manuscript has been proofread by American Journal Expert. See the certificate provided below.

[Figure]

**Editing Certificate**

This document certifies that the manuscript

**Submission to Soil (egusphere-2024-323)**

prepared by the authors

**Jeremy Rohmer, Stephane Belbeze, Dominique Guyonnet**

was edited for proper English language, grammar, punctuation, spelling, and overall style by one or more of the highly qualified native English speaking editors at AJE.

This certificate was issued on **June 25, 2024** and may be verified on the AJE website using the verification code **CD67-5F19-EADA-DD07-6CFD** .

[Figure]

Neither the research content nor the authors' intentions were altered in any way during the editing process. Documents receiving this certification should be English-ready for publication; however, the author has the ability to accept or reject our suggestions and changes. To verify the final AJE edited version, please visit our verification page at aje.com/certificate. If you have any questions or concerns about this edited document, please contact AJE at support@aje.com.

AJE provides a range of editing, translation, and manuscript services for researchers and publishers around the world. For more information about our company, services, and partner discounts, please visit aje.com.

**Replies to Referee #1's comments on "Insights into the prediction uncertainty of machine-learning-based digital soil mapping through a local attribution approach" (egusphere-2024-323)**

We would like to thank Referee #1 for the constructive comments. We agree with most of the suggestions and, therefore, we have modified the manuscript to take on board their comments. We recall the reviews and we reply to each of the comments in turn (outlined in blue). The main corrections made to the manuscript are described in a specific section of each response.

**Referee #1:**

*This is a review for the manuscript Insights into the prediction uncertainty of machine-learning-based digital soil mapping through a local attribution approach by Rohmer et al. The authors use SHAP, a common tool for assessing machine learning predictions at local scale, to investigate the contribution of covariates (or rather groups of covariates) on the uncertainty of a random forest model. It is well known that Shapley values are computationally very expensive, and so the authors propose to reduce the number of covariates to speed up computations. This is done before model training (a rather odd proposal) by using a statistical dependence test (i.e., HSIC), and then after model training by grouping covariates (again with the same dependence test). The main aim of investigating covariates with the model's uncertainty is intriguing within the field of digital soil mapping, but the manuscript has some major flaws. Major concerns are related to the methodology of the entire selection procedure of covariates as well as the with the presented case study. The quality of the writing is also unfortunately poor.*

*Main methodological concerns*

- *My first criticism is related to the first step, that is, the elimination of covariates before model training. This is a common pitfall within machine learning in DSM. The problem is with data leakage which may cause bias, and this occurred when covariates are removed from the entire training data set, and not within for example a cross-validation within each fold. Note that any data preprocessing (e.g., normalisation) dealt with in such a way can lead to data leakage. Data leakage may also cause the model's uncertainty to be lower, and this is then also problematic if interpretative machine learning (IML) methods (like SHAP) are used to analyse the relationships between covariates and the model's uncertainty. In addition, with a model such as random forest, covariate selection is not really required, especially with so few covariates (i.e., 15). I invite the authors to refer to the work such as that of Zhu et al. (2023) for guidance on data preparation so that data leakage is avoided.*

We are grateful to Referee #1 for pointing out the potential problem of data leakage. Now we better underline that the proposed screening analysis is conducted during the cross-validation procedure as recommended by Zhu et al. (2023). Our HSIC-based covariate selection is now analysed at each iteration of the cross-validation. Considering the 10-fold cross validation procedure (repeated 25 times), new Figure 10 shows the corresponding p values. The dots indicate the mean value estimated over the replicates of a 10-fold cross validation (repeated 25 times), and the lower and upper bounds of the error-bar are defined at +/- one standard

deviation. . When the dot merges with the error-bar, this indicates that the value of the standard deviation is low.

Covariates with p values below the 5% significance level are considered influential. This shows that, out of all the cross-validation replicates, nine covariates have a statistically significant influence on hydrocarbon concentration. These covariates are retained in the construction of the RF model.

[Figure]

**New Figure 8: Screening analysis showing the p values of the *HSIC*-based test of independence (described in Appendix C) for the Toulouse case. The dots indicate the mean values estimated over the replicates of a 10-fold cross-validation (repeated 25 times). The lower and upper bounds of the error bars are defined as +/- one standard deviation. When the dot merges with the error bar, the value of the standard deviation is low. The vertical red line indicates the significance threshold at 5%. When the p value is less than 5%, the null hypothesis should be rejected, i.e., the considered covariate has a significant influence on the hydrocarbon concentration and is retained in the RF construction.**

Regarding the usefulness of the screening analysis, we only partly agree with Referee #1's comment, because many real case studies have implemented such approaches to select covariates using either recursive feature elimination or forward feature selection or RF importance measures for cases with both very large number of covariates, such as the study by Poggio et al. (2021), but also with a moderate number of covariates of the order of 10-20 as in our case, such as the study by Meyer et al. (2019) and Dornik et al. (2022).

In our study, we rely on an alternative approach which has proven to be very efficient in the machine learning community (see e.g. Gretton et al., 2005) with applications in multiple domains, e.g. atmospheric pollution (Fellmann et al., 2023); environment (Lambert et al., 2024); geochronology (Herrando-Pérez & Saltré 2024); nuclear safety (Marrel and Chabridon, 2021); deep learning and image analysis (Novello et al., 2022); geothermics (Rohmer et al., 2023). The key advantages in our case are:
   (i)    *HSIC* measure can capture arbitrary dependence without resorting to some assumptions such as linearity, monotonicity;
   (ii)   *HSIC* measure can handle random variables potentially of mixed type, continuous or categorical;
   (iii)  *HSIC* measure avoids the use of RF importance measures, which show some limits as extensively discussed, among others, by Ishwaran (2007), Strobl et al., (2007),

Benard et al. (2022). This aspect has also been clearly underlined by Meyer et al. (2019).

The objective of the *HSIC*-based covariate selection is thus two-fold:
- Using a model-agnostic approach that avoids the use of an importance measure that is inherent to the selected ML model;
- Decrease the computational burden of the SHAP approach.

**Main correction:**
The covariate selection is now conducted in the cross-validation procedure following the recommendation of Zhu et al. (2023). The twofold objective of the *HSIC*-based covariate selection is now better highlighted in Sect. 3.1 'global procedure'. We recognise, however, that it is beyond the scope of this study to compare this method with alternative techniques available in the literature. This is indicated as a perspective.

**References**

Bénard, C., Da Veiga, S., & Scornet, E. (2022). Mean decrease accuracy for random forests: inconsistency, and a practical solution via the Sobol-MDA. Biometrika, 109(4), 881-900.

Dornik, A., Cheţan, M. A., Drăguţ, L., Dicu, D. D., & Iliuţă, A. (2022). Optimal scaling of predictors for digital mapping of soil properties. Geoderma, 405, 115453.

Fellmann, N., Pasquier, M., Blanchet-Scalliet, C., Helbert, C., Spagnol, A., & Sinoquet, D. (2023). Sensitivity analysis for sets: application to pollutant concentration maps. arXiv preprint arXiv:2311.16795.

Herrando-Pérez, S., & Saltré, F. (2024). Estimating extinction time using radiocarbon dates. Quaternary Geochronology, 79, 101489.

Ishwaran, H. (2007). Variable importance in binary regression trees and forests. Electronic Journal of Statistics, 1:519–537.

Lambert, G., Helbert, C., & Lauvernet, C. (2024). Quantization-based LHS for dependent inputs: application to sensitivity analysis of environmental models. https://ec-lyon.hal.science/hal-04546338/file/Article_QLHS.pdf

Marrel, A., & Chabridon, V. (2021). Statistical developments for target and conditional sensitivity analysis: application on safety studies for nuclear reactor. Reliability Engineering & System Safety, 214, 107711.

Meyer, H., Reudenbach, C., Wöllauer, S., & Nauss, T. (2019). Importance of spatial predictor variable selection in machine learning applications–Moving from data reproduction to spatial prediction. Ecological Modelling, 411, 108815.

Novello, P., Fel, T., & Vigouroux, D. (2022). Making sense of dependence: Efficient black-box explanations using dependence measure. Advances in Neural Information Processing Systems, 35, 4344-4357.

Rohmer, J., Armandine Les Landes, A., Loschetter, A., & Maragna, C. (2023). Fast prediction of aquifer thermal energy storage: a multicyclic metamodelling procedure. Computational Geosciences, 27(2), 223-243.

Strobl, C., Boulesteix, A.-L., Zeileis, A., & Hothorn, T. (2007). Bias in random forest variable importance measures: illustrations, sources and a solution. BMC Bioinformatics, 8:25.

- *Linking to my previous point. if the goal is to speed up computations, then removing covariates should not be a first choice. In addition, in typical DSM projects the number of covariates is usually more than 100. Therefore, the presented case study, which only has 15 covariates, is not the best choice to showcase the proposed methodology. One could rather perform a sample of grid cells at which Shapley values are estimated. Like for example in the Wadoux et al. (2023) paper. Again, in many DSM projects, maps are sometimes created over millions of grid cells, so the presented case study is not the best one to showcase this methodology. Therefore, to speed up computations with a small data set (like the one in this study), I would rather use a stronger machine to do the calculations than to omit potentially important parts of my data. If not possible, then let the computations run for a few days.*

We thank Referee #1 for the comments. It is true that the presented case study is not representative of very large scale projects with millions of grid cells and hundreds of covariates.

However, we would like to underline that numerous case studies have been found in the literature with number of covariates that are comparable to our case study. Among others, please refer to:

- de Bruin et al. (2022) used a set of 15-20 covariates to predict the organic carbon stock and the above ground biomass;
- Dornik et al. (2022) used 10-15 covariates to predict soil properties in Romania;
- Meyer et al. (2019) used 16 covariates to the classification of Land Use/Land Cover in Germany;
- Fendrich et al. (2024) used 17 covariates to predict arsenic in European topsoils;
- Milà et al. (2022) used 19 WorldClim bioclimatic variables for their synthetic case;
- Wadoux et al. (2023) used 23 covariates for their study.

Regarding the very large number of grid cells, we propose to improve the discussion on this aspect. In the discussion section, we now indicate how the proposed approach could be helpful for these challenging cases. The combination of the grouping and of the screening analysis allow us to decrease the computation cost from 1 day to less than half an hour given approximately 45,000 grid cells. Accounting now for the constraints of global scale studies such as Poggio et al. (2021), a direct SHAP analysis would imply >22 days of calculation, hence requiring a high performance computing architecture. Our approach would here imply <1 day of computation on a single laptop. Moreover, with the growing concern of energy consumption (see e.g., Jay et al., 2024) for scientific computing, we believe of the importance of providing the soil scientists with efficient, energy-saving analytical tools. The other side of the coin is however the introduction of some simplifications that are discussed in the reply to Referee #1's next comment.

**Main correction:**
While we believe that our actual case is representative in terms of number of covariates of many found in the literature, we recognise that our approach should be better discussed in relation to the challenge of large-scale projects. To this end, we have replaced section 5.2 'Added value of clustering' with a new section 5.2 'Large-scale implementation' to better highlight the challenge of handling hundreds of covariates as well as implementing studies on a national or global scale. This aspect is also highlighted as a perspective of the present study.

**References**

de Bruin, S., Brus, D. J., Heuvelink, G. B., van Ebbenhorst Tengbergen, T., and Wadoux, A. M. C.: Dealing with clustered samples for assessing map accuracy by cross-validation. Ecological Informatics, 69, 101665, 2022.
Dornik, A., Cheţan, M. A., Drăguţ, L., Dicu, D. D., & Iliuţă, A. (2022). Optimal scaling of predictors for digital mapping of soil properties. Geoderma, 405, 115453.
Fendrich, A. N., Van Eynde, E., Stasinopoulos, D. M., Rigby, R. A., Mezquita, F. Y., & Panagos, P. (2024). Modeling arsenic in European topsoils with a coupled semiparametric (GAMLSS-RF) model for censored data. Environment International, 108544.
Jay, C., Yu, Y., Crawford, I., Archer-Nicholls, S., James, P., Gledson, A., et al.: Prioritize environmental sustainability in use of AI and data science methods. Nature Geoscience, 1-3, 2024.
Milà, C., Mateu, J., Pebesma, E., & Meyer, H. (2022). Nearest neighbour distance matching Leave-One-Out Cross-Validation for map validation. Methods in Ecology and Evolution, 13(6), 1304-1316.
Meyer, H., Reudenbach, C., Wöllauer, S., & Nauss, T. (2019). Importance of spatial predictor variable selection in machine learning applications–Moving from data reproduction to spatial prediction. Ecological Modelling, 411, 108815.
Poggio, L., De Sousa, L. M., Batjes, N. H., Heuvelink, G., Kempen, B., Ribeiro, E., and Rossiter, D.: SoilGrids 2.0: producing soil information for the globe with quantified spatial uncertainty. Soil, 7(1), 217-240, 2021.

- *The grouping of covariates is a practical way of speeding up computation, but I am afraid it holds no meaning for DSM practitioners. The authors acknowledge this in the discussion,*

*starting at Line 519. Doing inference on machine learning output with IML methods is hard enough. I cannot see how the grouping of covariates could hold much interpretive meaning.*

From Referee #1's comment, we understand that we have falsely conveyed our message about grouping. It should not be understood as a "one-fits-all" method. Instead the grouping should preferably be used to help the interpretation of the Shapley values. This can be done by defining, from the beginning of the analysis, groups of covariates that hold:

- a certain redundancy in terms of information due to the strong dependency between them. This is the solution followed in the original version of our work, but we recognise that it cannot be the only one;
- a meaning for analysists or the end-users. This is the second grouping option by Jullum et al. (2021) based on underlying knowledge/expertise (i.e. grouping covariates that make sense with respect to the problem at hand). As a motivation for this option, the study of Wadoux et al. (2023) is illustrative. In the presentation of their results (Figure 6 of their study), they naturally propose to analyse groups of covariates.

**Main correction:**

In the revised version of the manuscript, we propose to:
- reformulate our message about grouping by presenting it as an option to facilitate the analysis, and not a mandatory step. The section 3.1 "overall procedure" has been reworked along these lines;
- re-analyse our case study by reworking the groups using the information on dependency and the experts' knowledge.

In our real case, we now analyse four groups of covariates:
- Land-use;
- The elevation.
- The group $D_{basias\text{-}water}$ which includes $D_{basias}$ and $D_{water}$. Since group reflects the general tendency of industrial sites to locate close to a water supply, the analyse of the joint influence is meaningful;
- The group of geographical coordinates, i.e. $D_{ne}$, $D_{se}$, $D_{nw}$, $D_{sw}$, and the $Y$-coordinate. This group of covariates were introduced to improve the predictive capability of the RF model by following the approach by Behrens et al. (2018). Interpreting the respective influence of each of these individual covariates is in practice tricky, and grouping them makes sense in this regard.

- *To sum up, exploring the relationship between covariates and model uncertainty is intriguing and worth exploring. However, the paper's emphasis on reducing computation with (questionable?) methods distracts from the main goal of the paper. That is, I would have liked to see more in-depth analysis of covariates related to SHAP (prediction) vs SHAP (uncertainty). I would also like to have seen more emphasis on: do we expect the same covariates to be related to both, why do we see different covariates in terms of predictions vs uncertainty.*

We agree with Referee #1 that our message on the methods has to be clarified.

**Main correction:**

We are now emphasising the advantages of the different stages (elimination of characteristics, grouping).

Particular attention has therefore been paid to
- Improving the implementation of the selection analysis based on the recommendation of Zhu et al (2023);
- Clarifying the definition of groups by combining information on dependency and expert knowledge;
- Improving the presentation of clustering as an option to facilitate the interpretation of Shapley values instead of a single method;
- Further developing the discussion regarding transfer to large-scale projects with millions of grid cells and hundreds of covariates.

We also agree with Referee #1 on the interest of further exploring the link between SHAP (prediction) vs SHAP (uncertainty). To improve this aspect, we propose to define a common level of comparison by normalizing the Shapley values with the same quantity, i.e. the predicted value. New Figure 10 has been updated accordingly (see below). This shows that the scaled Shapley's values are mainly in the range [0, 25%], but with some particular areas where the determining factors for one or other situation (best prediction estimate or uncertainty) are not necessarily the same.

[Figure]

**New Figure 10:** Scaled Shapley values (in %) for each group of covariates of the Toulouse test case considering the prediction best estimate using the RF conditional mean (left) and the prediction uncertainty using the qRF interquartile width (*IQW*) (right). The black squares indicate the locations of the soil samples used for RF training.

**Main correction:**

To deepen the analysis of the link between SHAP (prediction) vs SHAP (uncertainty), we propose to investigate in more details three distinct cases which are relevant from the viewpoint of uncertainty management.

- The first case corresponds to locations where at least one group of covariates contribute significantly to the uncertainty, by more than 25% compared with its contribution to the best estimate. This corresponds to <15% of study area;

- The second case is the opposite of the first one, and corresponds to locations where at least one group of covariates contribute significantly to the best estimate, by more than 25% compared with its contribution to the uncertainty. This corresponds to about 66% of the study area;
- Finally, the third situation overlaps with the second case and corresponds to where at least one covariates' group has negative contributions (in light blue in Figure 10, bottom), i.e. where they participate directly in reducing prediction uncertainty. This corresponds to a large proportion of the study area, of about 80%.

[Figure]

**New Figure 12: Boxplots of the covariate values for the training dataset and for the locations (named "selection") where the corresponding group of covariates contributes significantly to the best estimate, with a scaled Shapley value exceeding that of the uncertainty by more than 25%. The bottom right-hand panel compares the proportions of land use categories (AGR: agriculture, FOR: forests and grasslands, IND: industrial and commercial economic activities) for the selection and training datasets.**

As recommended by Referee #1, we analyse the analysis of the relationships by examining the distribution of the corresponding covariates. Comparison with the training data set gives us an insight into the reasons for the different situations. New Figure 12 illustrates the second case. It reveals that these locations have elevation values and distances $D_{basias}$ and $D_{water}$ of the same order of magnitude as those in the training dataset. This corresponds to a prediction situation where the RF model is used to predict cases that show similarities to the training dataset. This also means that this prediction situation does not rely too much on the extrapolation capability of the RF model; a situation known to be difficult for this type of ML model (Takoutsing and

Heuvelink, 2022). In the areas where land use contributes most to this case, we show (Fig. 12, bottom, right-hand panel) that this is linked to agricultural areas and forests, i.e. areas where there is less chance of finding potentially polluted sites, as shown by the analysis of the training dataset.

It should be noted that, for the sake of brevity of the revised manuscript, some of these analyses are included in the supplementary documents. In particular, the number of figures in the main text has been limited to around ten.

Reference
Takoutsing, B., and Heuvelink, G. B.: Comparing the prediction performance, uncertainty quantification and extrapolation potential of regression kriging and random forest while accounting for soil measurement errors. Geoderma, 428, 116192, 2022.

*Some other concerns / suggestions*

- *The synthetic case study adds no value to the paper. I suggest removing it as the paper is already a bit long for the topic at hand.*

We only partly agree with Referee 1' comment, because Referee 2 clearly emphasised the value of this synthetic case study in facilitating understanding of the methods. In addition, the data from the Toulouse case study has restricted access. We believe that having a synthetic test case that we can share publicly should facilitate the use and critical analysis of our approach. Therefore we have chosen to keep the synthetic case in the study.

- *Section 3.1 is difficult to follow without the knowledge of HSIC and some of the information in the many cited references. Maybe just restructure the manuscript and include essential methodology.*
- *Random forests are standard and already widely known in DSM. The sections on RF and QRF can be removed, and replaced with brief references to RF and QRF.*

We reply here to both comments. To improve the clarity of the presentation, we have reformulated Sect. 3 by:
- describing the essential details of the methodology in a section Sect. 3.1 "overall procedure";
- describing in full details in section Sect. 3.2 "3.2 Shapley additive explanation" the approach based on Shapley values with additional comments on the computational burden and the benefit of grouping;
- moving the sections on RF models and on HSIC dependence measure in Supplementary Materials.

- *Maps presented in this manuscript are of poor quality and not visually appealing. Captions and legend can also be improved. With Figure 3, show more information. Not everyone is that familiar with this region in France. The histogram is not very clear, especially the long right tail can be enhanced visually.*

The maps have been reworked. In particular the locations of the samples are systematically added to the maps to ease the interpretation of the results with respect to the training dataset. More appealing color scales have been chosen, namely "Set3" and "Paired" from ColorBrewer (https://colorbrewer2.org/); see above an example with new Figure 10. The captions and legends have also been further detailed.

In particular, the presentation of Figure 3 has been improved.

[Figure]

**New Figure 3: (a)** Spatial location of the 1,043 soil samples (square-like markers) across Toulouse city located in the South-West of France (see the grey dot in the top right inserted map). The size of the squares is proportional to the logarithm (base 10) of the C10-C4 hydrocarbon concentration (expressed in mg/kg). **(b)** Histogram of the logarithm (base 10) of the C10-C4 hydrocarbon concentration (expressed in mg/kg) with a zoom on the interval 2.0-4.0 (top right inserted panel).

- *General writing of the manuscript is poor. Some examples: The overuse of "etc", too many brackets to give additional information, brief introductions at each section.*

Careful rewriting has been carried out to avoid unnecessary 'etc' and the brackets. The brief introductions have been removed.

- *The mathematical writing can also be improved. For example, are the ahuthors sure that ML model is just y=f(x)? See Line 142.*

The mathematical writing has been cross-checked and the identified problem in line 142 has been corrected as follows: "The mathematical relationship is modelled by a ML model (denoted $f(.)$) sot that $f(\mathbf{x}(s))$ is assumed to resemble $y(s)$ as closely as possible. i.e. $y(s) \approx f(\mathbf{x}(s))$."

- *Figure 6 does not make sense. Why is there an arrow from Step 2 to 4?*

From Referee #1's comment, we have the impression that Figure 6 introduces some confusion. We propose to remove it and to describe in more details the different steps and their interplay in the sub-section Sect. 3.1 "overall procedure".

*References:*

*Wadoux, A., Saby, N., Martin, M. (2023). Shapley values reveal the drivers of soil organic carbon stock prediction. SOIL, 9, 21-38. doi: 10.5194/soil-9-21-2023.*

*Zhu et al. (2023). Machine Learning in Environmental Research: Common Pitfalls and Best Practices. https://pubs.acs.org/doi/10.1021/acs.est.3c00026.*

**Replies to Referee #2's comments on "Insights into the prediction uncertainty of machine-learning-based digital soil mapping through a local attribution approach" (egusphere-2024-323)**

We would like to thank Referee #2 for the positive analysis and the constructive comments. We agree with most of the suggestions and, therefore, we have modified the manuscript to take on board their comments. We recall the reviews and we reply to each of the comments in turn.

**Referee #2:**

*This manuscript is well written, clear and relevant, and presents methods that could provide stakeholders with valuable insights into where the uncertainty comes from: this has the potential to make uncertainty more concrete for them.*
*I appreciate the use of a synthetic test case, which makes the whole procedure a lot easier to understand.*
We thank Referee #2 for this positive feedback.

*I don't have any major criticisms. I would be pleased to see this manuscript published after attention to the following minor details :*

*Line 44: However, at a local scale, these methods don't (?) provide any information for a prediction at a certain spatial location.*
We thank Referee #2 for noticing this problem. We have reformulated as follows: "However, these methods do not allow to measure the influence of the covariates for a prediction at a certain spatial location."

*Line 157: pushes the prediction uncertainty?*
We agree that this term is confusing. The sentence has been replaced as follows: "i.e. whether the considered covariate influences the prediction upwards or downwards in relation to the base value"

*Line 442: I don't see any circular pattern on the bottom middle panel of Figure 13 (in the bottom right one however, they are really clear).*
We thank Referee #2 for noticing this problem. We have corrected the text by referring to Figure 13, bottom right.

*Synthetic test case: isn't the fact that in Z1, the biggest contributor to uncertainty is Tmean-Tmax (and that respectively in Z2, the biggest contributor is Pwettest) be linked to the fact that these covariates have uniquely high (respectively low) values there, that are not represented in the dataset? If you agree, this in my opinion would be interesting to put in the discussion.*
We thank Referee #2 for the analysis. These additional elements are helpful for a better understating of the synthetic case and have been added to Sect. 4.1.

Orleans,
June 25[th], 2024
J. Rohmer[1] on behalf of the co-authors

[1] BRGM, 3 av. C. Guillemin - 45060 Orléans Cedex 2 – France

---

## Author Response (AR2)

**Replies to Referee 1's comments on "Insights into the prediction uncertainty of machine learning-based digital soil mapping through a local attribution approach" (egusphere-2024-323)**

We would like to thank the Editor and Referee 1 for giving us the opportunity to correct our manuscript based on the new comments. We agree with most of the suggestions and, therefore, we have modified the manuscript to take on board their comments. Marked changes are indicated in green. The line numbers are those of the manuscript with tracked changes.

**Editor:**

*R1 has noted improvement in the revised manuscript but also considered that some more changes are needed before we can accept the manuscript. I agree with this report and hope that authors can answer the remaining comments.*

We thank the Editor for this new round of review. We have paid a special care to address the problems raised by Referee 1 by clarifying multiple aspects; in particular the problem of spatial clustering, and of spatial extrapolation, as well some confusing terms.

In the acknowledgement section, we also indicate that we are grateful to the two anonymous referees for their comments and recommendations that led to the improvements of the manuscript.

**Referee 1:**

*I commend the authors on an improved manuscript. The methodology section now reads better, and the various steps are also clearer. I am also pleased with the improved methodology in terms of performing the screening analysis within the cross-validation. However, even with the improvements, certain sections are still a bit cryptic. See additional concerns below (some are minor and other are more major). I based my second review on the track changes document.*

*Line 260. The sentence starting with "Overall, the RF …" reads strange.*
The term "Overall" was removed.

*Line 280. maybe rather: "Shapley values, as defined in Sect. 3.2, … ". Avoid to overly depend on brackets.*
This is now corrected.

*Line 289-291. Try rewriting in more than one sentence. It is currently hard to follow. In addition, I am unsure what the authors mean by "… having uniquely high and low values…".*
We agree that this sentence is unclear. We now more clearly indicate the problem of representativeness of the training dataset as follows: "These differences in importance may be related to the scarcity of soil samples in both zones (see Fig. 2). This means that the training data are not representative of both zones".

*Line 299-300. First part of the sentence does not make sense. The part with "to estimate the conditional mean, which is used as the best estimate of the prediction,…". Rewrite, because this is technically wrong. How can the estimate of the conditional mean be the estimate of the prediction?*

The sentence is now rewritten as follows: "We use the conditional mean as the best estimate of the prediction, and the interquartile width *IQW* as the uncertainty estimate, with the 25th and 75th quantiles computed using a qRF model."

*Line 301. I must admit I am getting lost with this part. Maybe other readers will as well. The authors mention the difficulty of related to the clustering (i.e., verb) of the observations. Because this term is also used in this manuscript to refer to the clustering algorithm, this is a bad choice for meaning how the points are distributed. Could the authors clarify what they mean here. Are they referring to how the points are spatially distributed?*

We thank Referee 1 for noticing this problem. We confirm that Referee 1's interpretation is correct. The problem is related to how the points are spatially distributed.

*Line 301: I am also confused as to why this is a problem? Given that my understanding of the above point is right.*
*Lines 302-308. Is all of this necessary? Was this discuss in the methodology section? So, to make sure I understand all of this. Since the points are spatially clustered, that is, the points are not well dispersed over the region, the authors define weights which must then be used when observations are sampled when the bootstrap samples (i.e. trees) are drawn. If my understanding is correct, then this seems all a bit unnecessary. Could the authors elaborate why this is necessary?*
*In addition, why would you bring additional methodology that was not discussed in the previous sections?*
*Also, what if the weights do not address the feature space well?*
*Another question, is this step necessary when you include covariates that used to address the spatial aspect of the data? I mean, you included covariates such as the coordinates and various distances.*
*Can the authors highlight DSM studies where this has been done? Again, I am just trying to understand the motivation behind this methodology in these lines.*

We reply below to this series of comments.

In order to clarify why this is a problem, we provide more details in Sect. 4.2 (page 14, lines 300-304) as follows: "In our case, one additional difficulty is related to how the points are spatially distributed. Figure 3a shows that the points are spatially clustered as they overrepresent some regions while underrepresent, or even miss, others. This situation might lead to biased predictions, because the same weight is given to every point and thus regions with high sampling density are overweighted."

It is important to note that in the original version of the study, we overlooked this problem. At the first submission stage, the Editor rightly pointed out the need to remedy it. We have therefore adopted a weighting procedure adapted from Bel et al. (2009) and Xu et al. (2016) that implies weighting the training data by the inverse sampling intensity.

We preferably describe the approach in section 4.2, because this problem is specific to our real case and because we do not claim to provide a new approach to solving it (see our last answer below).

Adopting the weighting procedure had two implications:
- a clear decrease of the prediction uncertainty, and an intensification of the prediction uncertainty in the zones where the initial RF (without weighting) was not "confident";
- the emergence of some clearer spatial patterns.

As rightly pointing out by Referee 1, the proposed approach using sampling intensity can be improved in different manners. Estimating the weights using the feature space is certainly a valuable suggestion. Some preliminary investigations have been tested in a conference (see e.g., https://geostat23.sciencesconf.org/489353). Alternative options have also been proposed in the literature; see e.g., the reference to de Bruin et al. (2022), which provides a recent example of DSM study having addressed this spatial clustering problem.

It is important to note that our objective in this study goes beyond improving the predictive capability of the RF model. Given the level of prediction uncertainty obtained using "classical methods", we aim to investigate what are the main drivers of this uncertainty and whether they differ from the ones driving the best estimate of the prediction. Therefore we have highlighted in the concluding remarks (page 24, lines 512-516) some lines of improvements on that particular topic as well on others as follows: "Third, we focused on one type of machine learning model, i.e., the quantile RF model. Alternative approaches should be considered in future research: different types of machine learning models, such as deep learning techniques, which have shown promising results (see, e.g., Kirkwood et al. (2022)), and improved approaches, in particular, to address complex sample distributions such as clustering (see, e.g., de Bruin et al. (2022) and references therein)".

**Reference:**
De Bruin, S., Brus, D. J., Heuvelink, G. B., van Ebbenhorst Tengbergen, T., and Wadoux, A. M. C.: Dealing with clustered samples for assessing map accuracy by cross-validation. Ecological Informatics, 69, 101665, 2022.

*Line 337: "...covariates are retained in the construction of the RF model." But the RF was already constructed if the cross-validation was performed. So why are covariates retained? What does this mean?*
*Line: 361. Oh, I see retained for the group based shap. Is this what the authors meant at Line 337? If so, then make it clearer. If not, please explain.*
Referee 1 is correct. Thank you for noticing this confusing statement. We now specifically specify in Sect. 4.2 (page 15, line 340) that: "These covariates are retained in the construction of the final RF model that is used for the application of the group-based SHAP approach."

*Line 406: models, plural?*
This typo is now corrected.

*Line 406: This is also a very strange sentence, because the RF model cannot extrapolate. See this post for example that explains it (https://stats.stackexchange.com/questions/235189/random-forest-regression-not-predicting-higher-than-training-data#:~:text=Decision%20Trees%20%2F%20Random%20Forrest%20cannot,outside%20of %20the%20observed%20range. ). So again, all of this is a bit cryptic, and I am cautious to what the authors mean (Lines 405-412). The authors referenced here the paper by Takoutsing*

*and Heuvelink. Note the paragraph right above section 3.5 that also notes that RF cannot extrapolate beyond training data.*
*L411: What limitations?*
We reply to the two comments (L406 and L411). We now clarify the term "extrapolation" as follows: "This indicates that the RF model is being used here beyond the area from which the training data were taken. This is a situation of spatial extrapolations, where tree-based methods such as RF can fail completely; see a recent study highlighting the limitations by Takoutsing and Heuvelink, (2022)".

We also clarify line 396 in Sect. 3.5 as follows: "[…] resulting in an "optimal" prediction situation in which the RF model is used to predict cases that are relatively similar to those used for its training".

*Lines 438-443: rewrite to include the long line-in reference in the quotes.*
Both sentences are now grouped in a unique one as follows: "The SHAP results are expected to improve the framing of the prediction results together with the associated uncertainty as illustrated with the synthetic test case described in the introduction as follows: […]".

*Line 456: extrapolation mode: odd way of stating that RF is used to make spatial extrapolations. See also in Line 411.*
To avoid confusion, we now use the term "spatial extrapolation" as suggested by Referee 1.

Orleans,
August 12[th], 2024
J. Rohmer[1] on behalf of the co-authors

[1] BRGM, 3 av. C. Guillemin - 45060 Orléans Cedex 2 – France